# Difference in gaze control ability between low and high skill players of a real-time strategy game in esports

Inhyeok Jeong[1], Kento Nakagawa[2]*, Rieko Osu[3], Kazuyuki Kanosue[2]

**1** Graduate School of Sport Sciences, Waseda University, Saitama, Japan, **2** Faculty of Sport Sciences, Waseda University, Saitama, Japan, **3** Faculty of Human Sciences, Waseda University, Saitama, Japan

* nakagawa.kento.22@gmail.com

## Abstract

This research investigated the difference in aspects of gaze control between esports experts (Expert) and players with lower skills (Low Skill) while playing the real-time strategy game called StarCraft. Three versions of this game at different difficulty levels were made with the StarCraft Editor, and the gaze movements of seven Expert and nine Low Skill players were analyzed while they played the games. The gaze of Expert players covered a significantly larger area in the horizontal direction than the gaze of Low Skill players. Furthermore, the magnitude and number of saccadic eye movements were greater, and saccade velocity was faster in the Expert than in the Low Skill players. In conclusion, StarCraft experts have a specific gaze control ability that enables them to quickly and widely take visual information from all over the monitor. This could be one of the factors enabling StarCraft experts to perform better than players with lower skills when playing games that require task-switching ability.

## Introduction

With the proliferation of personal computers, smartphones, and the internet, many people have been able to easily enjoy playing video games. Today it is common to play a competitive game with other players online; this activity is called electronic sports (esports) [1]. Irrespective of their genre, esports are very competitive and often professionally played. According to research into the economics of esports, the total value of global esports competition prizes increased from $360 million in 2005 to $7.1 billion in 2015 [2]. The expansion of esports markets provides more chances for top level esports players to earn large amounts of money. In addition, meta-analysis revealed that playing a commercial video game improves the information-processing skills of players, which include task-switching ability and visual processing [1, 3]. Task-switching is defined as the ability to quickly alternate among multiple separate tasks [4]. To be specific, Kowal et al. [5] found that the task-switching ability of action video game (AVG) players is higher than that of non-AVG players. Furthermore, top video game players have better cognitive flexibility than novice players [6] in their ability to adapt to new and unexpected conditions [7]. In addition, a recent meta-analysis also found that video game

**Data Availability Statement:** All relevant data are within the manuscript and its Supporting information files.

**Funding:** This work was supported by JSPS KAKENHI Grant Number 18H04087.

**Competing interests:** The author received no specific funding for this work.

practice has positive effects on cognitive function [8] (the mental processing required to obtain information and knowledge [9]).

The real-time strategy (RTS) game is a popular esports genre. In an RTS game, players are required to task-switch using complex strategies, because multiple information streams appear simultaneously in different locations on the monitor screen, and players must select and absorb information of importance to them [10]. Based on the information obtained, they must judge how to make an appropriately timed response to the stimuli. It is probable that considering these characteristics of an RTS game, positive effects of playing RTS games on cognitive processes (reaction time and problem-solving ability) occur [11]. Dobrowolski et al. [12] found that RTS players have superior cognitive function compared to that of AVG players. Training adults older than 60 to play RTS games improved their task-switching [13]. This could be accompanied by or be based on improving the gaze control ability which RTS requires; that is, for smooth task-switching, it is necessary to quickly process visual stimulation from multiple sources.

In light of the above background, we used a popular RTS game, StarCraft, as a research model of RTS, and aimed to clarify how StarCraft experts control their gaze during their play to estimate their gaze control ability. Since task difficulty (i.e., number of tasks that players have to switch between) would be a critical factor for gaze control, it is necessary to clarify whether or not gaze control depends on task difficulty. Thus, we set the test games with three different levels of difficulty. Additionally, it is already known that first-person shooting (FPS) games improve task-switching ability [14]; in addition, not only a single case study but also a meta-analysis study argues that RTS improves task-switching ability [11]. One study has found that FPS games and RTS games have a similar positive effect on task-switching ability [15]. Thus, elite StarCraft players should have superior task-switching ability. Furthermore, as a source of their superior task-switching ability, they would use more ballistic gaze movements, called saccades, to quickly process multiple stimuli. We hypothesized that StarCraft experts would show specific gaze control (dispersed and fast saccadic gaze movements) while playing games compared to the novices. We tested this hypothesis by measuring gaze distribution and saccadic movements.

## Methods

In the present study StarCraft was utilized as a model game to study task-switching ability. StarCraft is a strategy simulation game developed by Blizzard Entertainment in 1999 (https://starcraft.com/en-us/). After the original was released in 1999, "StarCraft: Remastered" featuring some graphics changes was released in 2017.

We performed a power analysis to estimate the required sample size (G*Power version 3.1). The power analysis was conducted using the Hard Task performance level (described in the Sample games section) of experts (Expert, n = 4) and the performance of players with lower skills (Low Skill, n = 4) in a preliminary experiment (Cohen's d: 1.91; α level: 0.05; power (1-β error probability): 0.9). Effect size was calculated according to Cohen [16]. The results showed that seven participants were required for each group. The details of the G*Power protocol are presented in the S6 File. As a result, sixteen participants (15 male, 1 female; 7 Expert, 9 Low Skill; mean age, 22.4 years; age range, 18–28 years) with experience playing StarCraft participated in the present study. Participants were recruited through the Waseda University school bulletin board. Subjects had no record of visual disorders, and were excluded if they had less than six months experience playing StarCraft. Subjects were divided into two groups, the Expert group and the Low Skill group, according to the history and official ranking of StarCraft game players. Expert is defined as those who have played StarCraft more than three

times a week for at least six months, or who are in the top 10% of the official StarCraft ranking (rankings by Blizzard Entertainment, developer of StarCraft). Low Skill players had not played StarCraft for more than six months, or their official ranking was in the bottom 50% of players. Before the experiment, we verbally provided information about the contents and concepts of this research along with the instruction documents. After that, we obtained verbal informed consent from all subjects. The research was approved by the Human Research Ethics Committee of Waseda University, Japan (2019–342).

### Experimental procedure

Before the experiment, we measured a distance of 40 inches from the monitor and asked subjects to maintain their heads at a 40 inch distance from the monitor during the task. Each task was performed for 3 minutes. When the participant failed to play the task for 3 minutes, the task was restarted at 0 minutes. Any task which was not played for 3 minutes was excluded from the analysis. During each task, gaze movement was recorded by an eye tracker (Pupil Core, Pupil Labs). The tasks proceeded in the following order: Easy Task, Moderate Task, and Hard Task (Fig 1). When a task had been performed for 3 minutes, the task was over.

### Sample games

The screen on which different information about the StarCraft areas was displayed is shown in Fig 2. In the game the player must construct as many buildings and produce as many product units (described later) as possible from resources (virtual commodities), and at the same time escape from and destroy enemies. In this process a high level of task-switching ability is required to perform all these jobs simultaneously.

Four different types of unit exist; commander unit, enemy unit, labor unit, and product unit. There are three Zones in which different jobs are performed. Specific features of units and locations of Zones are shown in Fig 3. Zone 1: one commander unit controlled by the player and three enemy units exist. Enemy units are avatars which are operated by the computer system. The commander unit fights with or escapes from enemy units that attack the commander unit. When all three enemy units are destroyed, three new enemy units reappear in Zone 1. Zone 2: labor units, product units, and buildings exist. Zone 2 is used to collect

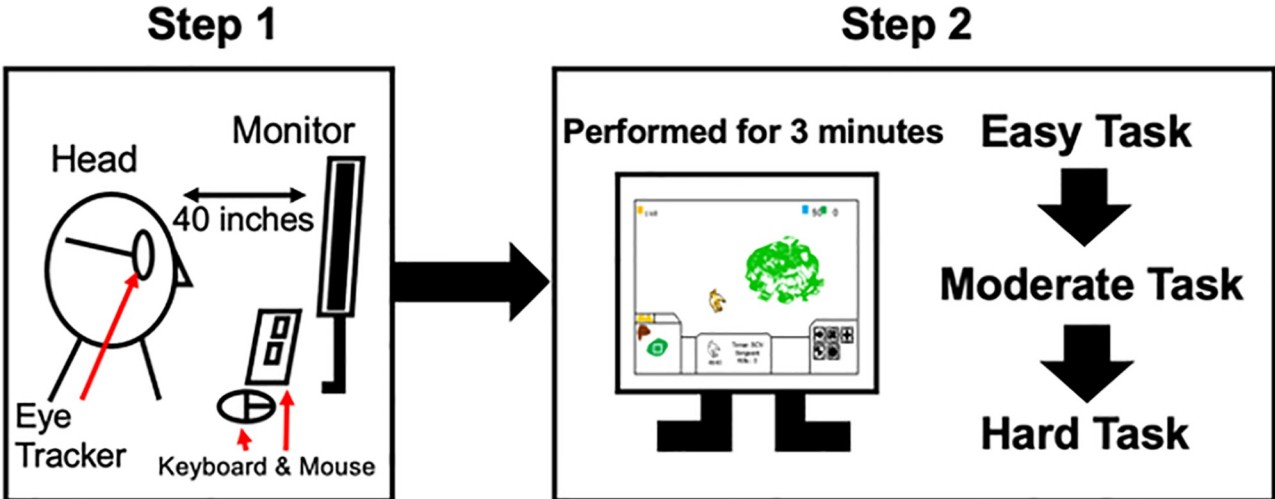

**Fig 1. Overall flow of the task.** These steps show the overall flow of the task.

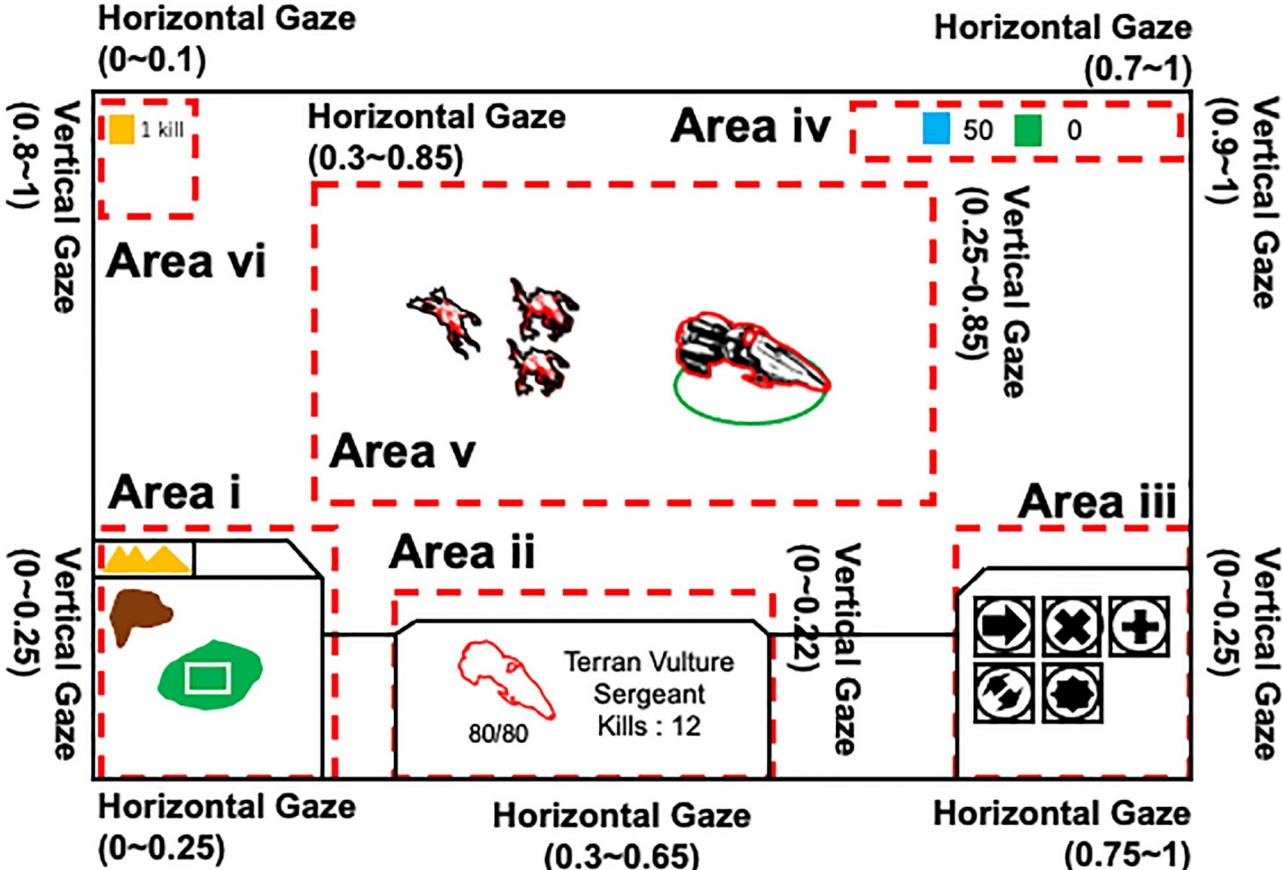

**Fig 2. StarCraft main play screen.** Dashed lines indicate six areas. Values in parentheses show the relative coordinates of each area normalized to monitor size, 0 to 1 in both horizontal and vertical directions. Area i: a mini-map which shows a bird's-eye view of all play Zones. When a critical event such as an attack from the enemy occurs in a certain Zone, the Zone blinks to inform the player that an event has occurred. Area ii: information about the number of destroyed enemy units, and remaining strength of the commander unit which is operated by the player. Area iii: the function(s) of a selected unit or building. Area iv: the number of resources. Area v: the play area which is a part of the selected Zone. Area vi: the total score amassed thus far by destroying enemy units. The various kinds of units, buildings, and Zones are described in the main text.

resources, construct buildings, and produce product units. Though there are multiple labor units, they move together as a group to collect resources, and to construct buildings. Once the game starts, the group of labor units automatically collects resources at a constant rate. Product units are produced by the player, consuming resources. Zone 3: buildings and product units exist. Zone 3 is used only to produce product units. All units, including individual units (commander, ememy, product) and the group of labor units, can move in any direction but cannot get out of their zones. The player must accomplish necessary jobs by switching the Zone that appears in Area v.

For the present study, three sample games (called tasks) with different levels of difficulty, Easy Task, Moderate Task, and Hard Task, were programmed using the StarCraft Campaign Editor (Blizzard Entertainment., U.S.A) (Fig 3). Example of play scenes in each task can be seen in the S1, S2 and S3 Files. The difficulty of each task depended on how many jobs were required and how many Zones were in play at the same time. When more jobs were required and more Zones were active at the same time, the difficulty of the task increased.

In the Easy Task, the player must accomplish three jobs; enable the commander unit to escape from enemy units in Zone 1, while at the same time in Zone 2 producing product units

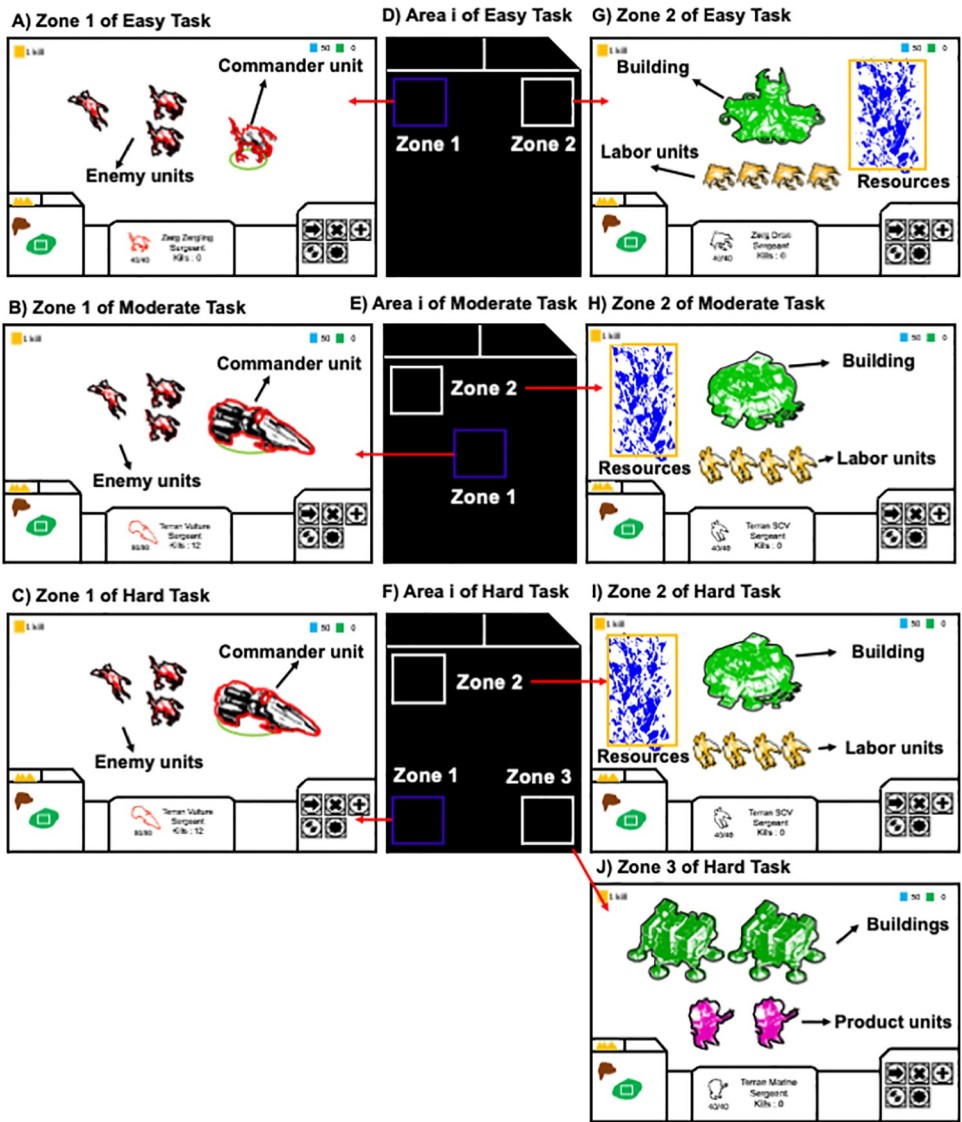

**Fig 3. Three different types of task-switching activities.** The meaning of each box is as follows. white boxes: Collecting resources and constructing buildings are done in these zones (Zone 2 and Zone 3). Blue box: The battle between the commander unit and enemy units occurs in this zone (Zone 1). Yellow box: Resources.

and constructing buildings from resources collected by four labor units. In the Moderate Task, the player can destroy enemy units in addition to completing the three jobs of the Easy Task. To destroy three enemy units, the commander unit must land two bombs on each enemy unit, for a total of six bombs used to destroy them all. In the Hard Task, the player was further required to produce product units in Zone 3. There were six buildings in Zone 3 at the start, and product units were produced in those buildings by the player. In Zone 2, 12 labor units collected resources at a faster rate than the four labor units in the Easy and Moderate Tasks. Zones 2 and 3 were synchronized so that resources did not have to be moved from Zone 2 to Zone 3. The positions of the three Zones differ among the three tasks (Fig 3D, 3E and 3F).

Once the player starts collecting resources, they continuously increase at a rate of 32 resources every 3 sec in the Easy and the Moderate Tasks, and 32 resources every 1 sec in the

**Table 1. Operation method.**

| Operation | Method |
|---|---|
| Select a unit or building | Mouse left click |
| Escape from enemy units | Select the commander unit + mouse right click |
| Switch Zones | "F2, F3, F4, F5" on the keyboard or "Spacebar" on the keyboard |
| Switch the part of a Zone shown in Area v | "Arrow keys" on the keyboard or move "Mouse cursor" |
| Destroy enemy units | Select the commander unit + "P" on the keyboard + mouse right click |
| Collect resources | Select the labor unit + mouse left click on resources (this is done only once when the task starts, after which resources are increased automatically) |
| Produce product units | Select the building + mouse left click on building |
| Construct a building | Select the labor unit + mouse left click on Area iii |

Hard Task. There is no limit on the number of buildings and product units produced. Operations utilized in the tasks are listed in Table 1 and the S4 File.

A task was played for three minutes. When the task did not last for three minutes because a player lost the game, the player repeated the task until they had played for a total of three minutes. The tasks proceeded in the following order: Easy Task, Moderate Task, and Hard Task. The health point (HP) is defined as the strength of the commander unit against an attack by enemy units. A game ended when the HP became zero due to attack (1st condition causing the game to terminate, the player's loss). The HP decreased by five every time the commander unit was contacted by enemy units. In the Easy Task, the HP started at 35 and decreased by five with each enemy contact, while at the same time automatically increasing by 0.75 per second (the maximum HP is 35). In the Moderate Task and the Hard Task, the HP started at 75. There was no automatic increase of the HP; it recovered to the original level when the commander unit destroyed three enemy units (the maximum amount to which the HP can recover is 75). The number of resources started at 50; if resources were not used they increased by 32 every 3 sec in the Easy and Moderate Tasks, and by 32 every 1 sec in the Hard Task. If the amount of resources reached 450, the game was over (2nd condition causing the game to terminate, player's loss). Thus, participants must use resources, by producing product units or constructing buildings, to ensure that the amount of resources does not reach 450. The above mentioned conditions for game over are summarized in Table 2.

**Table 2. Three types of task and common game-over conditions.**

| Task | Details |
|---|---|
| A) Easy Task | Jobs are done in two different Zones at the same time. |
|  | Zone 1: escape from enemies |
|  | Zone 2: collect resources + produce product units + construct buildings |
| B) Moderate Task | Jobs are done in two different Zones at the same time. |
|  | Zone 1: escape from enemies + destroy enemy units |
|  | Zone 2: collect resources + produce product units + construct buildings |
| C) Hard Task | Jobs are done in three different Zones at the same time. |
|  | Zone 1: escape from enemies + destroy enemy units |
|  | Zone 2: collect resources + produce product units + construct buildings |
|  | Zone 3: produce product units |
| Common game-over conditions | Amount of resources (minerals, gas) becomes greater than 450 (loss). |
|  | HP of the commander unit becomes 0 (loss). |

All participants in the preliminary experiment agreed that a high level of task-switching ability was required during the task.

## Behavioral data analyses

To compare Expert and Low Skill players in their performance of the Easy Task, we defined the score of behavioral performance as:

$$(number\ of\ constructed\ buildings + number\ of\ produced\ units),$$

For the Moderate Task and the Hard Task, the score was defined as:

$$(number\ of\ constructed\ buildings + number\ of\ produced\ units) \times 2 + (number\ of\ destroyed\ enemy\ units) \times 3$$

The reason for using different constants is to strongly emphasize the number of destroyed enemy units. In order to compare the number of clicks between the Expert and Low Skill players, we measured the number of key presses and mouse clicks per minute, and defined this number as Actions Per Minute (APM). APM were automatically counted in the StarCraft program.

## Equipment

The experiment utilized 24–34 inch computer monitors. Participants selected the monitor they wanted to use. There was no significant difference in the selected monitor size between Expert players (average of monitor size: 27.00 inch ± SD 3.87) and Low Skill players (average of monitor size: 28.77 inch ± SD 4.35) (Wilcoxon Rank-Sum test, $p$ = .30). Before the task started, we measure the distance (40 inches) between each participant's head and their monitor. After measuring the distance, we instructed participants to keep the position of their head steady. Participants used their own keyboard and mouse.

Gaze movement was measured by the Pupil Labs eye tracker (Pupil Labs UG haftungsbeschränkt, Berlin, Germany). The open source software called Pupil-Capture version 1.15 was used for measurement (https://github.com/pupil-labs/pupil/releases/tag/v1.15). Gaze movement was measured by one field camera (60Hz @ 1910 × 1080 pixels) and two eye cameras (200Hz @ 192 × 192 pixels). The field camera was used to record the locations of the calibration markers. The screen marker calibration method was used to measure gaze movement (see screen marker calibration [17]). In addition, four surface markers (4cm × 4cm, height × width) were attached to each corner of the monitor. These markers were used to assign X and Y coordinates to the horizontal and vertical gaze positions. The coordinates of all gaze movements on the monitor were normalized to the width and height of that particular monitor and expressed with values between 0 and 1.

## Gaze signal analyses

Measured eye movement data were analyzed using Pupil-Player (Pupils Labs, v 1.15). The Pupil Labs guidelines recommend that detection data be used only when the confidence value of the pupil's center location is more than 60% (https://docs.pupil-labs.com/core/software/pupil-player/#raw-data-exporter). In this experiment, all gaze data with confidence value less than 70% were excluded (total ratio of excluded data: 6.92%) to improve analysis reliability. Accuracy was calculated as the average angular offset (distance) (in degrees of visual angle) between fixation location and the corresponding location of the fixation target [17]. All gaze movement data were normalized to a value between 0 and 1 with the Pupil-Player Surface Tracker software using the monitor size (mm) of each participant, because participants used different sized monitors. Standard deviations of gaze distribution in horizontal and vertical

directions (SD of gaze distribution) were separately calculated to compare the distributions of gaze position.

## Classification of gaze movement

The eye movement detector in the Pupil-Player (Pupil-Labs, v 1.15) software was applied to classify the type of gaze movement into two categories (saccade, fixation) based on a linear regression curve obtained by gaze movement segmentation [18]. The word saccade describes fast ballistic eye movements which radically change visual input on the retina [19]. When the eyes remain stable at a point it is called fixation [20]; fixation helps the eyes align with the target and avoid perceptual fading [21]. The classification was based on the Hidden Markov model (I-HMM), which classifies gaze movement using this probabilistic model (I-HMM is not based on velocity threshold and duration time) [18, 22]. In this identification method, two probabilistic models within I-HMM classify gaze movement: observation probabilities and transition probabilities. When the expected velocity of a gaze movement is high, the observation probabilistic model defines the movement as saccade. When the expected velocity is low, the movement is defined as fixation. The transition probabilities model calculates the probability of conversion between saccade and fixation. After classification, the numbers of saccade and fixation events were calculated as percentage of all gaze movements (saccade percentage, fixation percentage). To compare the characteristics of saccade between Expert and Low Skill players, velocity, length, and number of saccadic movements were analyzed.

We investigated the ratio of fixation in each area (Area of Interest; AOI); that is, the areas on the monitor that subjects looked at during task execution. To calculate the ratio, fixations in an area were extracted from the gaze movement, and the ratio was obtained as the summed time of the fixations in the area divided by the total fixation time in all areas in the three minutes of total task execution.

## Statistical analyses

All statistical analyses were conducted using RStudio version 1.3 (RStudio, Boston, MA). Homogeneity of variance was tested with Levene's test and normality of variance was tested with Shapior-Wilk's test. As a result, SD of gaze distribution (vertical gaze), ratio of fixation in each area (AOI), and saccade velocity did not follow a normal distribution of variance and show homoscedasticity. For this reason, these data were analyzed by a non-parametric method. When the Kruskal-Wallis test detected significance, the Wilcoxon Rank-Sum test was performed to determine which task difficulties were significantly different from one another (Bonferroni's correction [$p < .05$ divided by 3 tests: significance threshold at .017]).

Comparison of behavioral data and APM between Expert and Low Skill players was conducted by an unpaired t-test at each level of difficulty (Easy, Moderate, and Hard). SD of horizontal gaze distribution, type of gaze movement (saccade or fixation) as classified by the I-HMM, and two saccade characteristics (saccade number, saccade length) were analyzed by 2 (skill level: Expert, Low Skill) × 3 (task difficulty: Easy Task, Moderate Task, Hard Task) two-way repeated measures ANOVA. When a significant main effect of task difficulty was found, simple main effect analyses were used to check difference in task difficulty. When significant interaction was found, Tukey's Honestly Significant Difference (HSD) post-hoc analysis was performed. Partial $\eta^2$ indicated effect size for the ANOVA. All statistical significance was set at $p < .05$.

## Results

### Behavioral data

There was no significant difference in Easy Task performance score between Expert (mean 15.00 ± SD 2.64) and Low Skill (mean 10.11 ± SD 3.72) players. For the Moderate Task and the Hard Task, an unpaired t-test revealed that the performance level of Expert was significantly higher than that of Low Skill players (Moderate Task: mean 170.28 ± SD 19.20 vs. mean 137.22 ± SD 25.97, t = 3.31, df = 15.89, $p$ = .004, and Hard Task: mean 208.14 ± SD 26.01 vs. mean 155.55 ± SD 37.26, t = 3.75, df = 15.71, $p$ = .001). For APM, there was no significant difference between Expert and Low Skill players in all tasks (Easy Task: mean 232.42 ± SD 42.42 vs. mean 212.77 ± SD 61.85, t = 1.06, df = 13, $p$ = .30, Moderate Task: mean 257.28 ± SD 69.16 vs. mean 267.00 ± SD 42.01, t = -0.03, df = 7.55, $p$ = .97, and Hard Task: mean 260.42 ± SD 54.70 vs. mean 256.33 ± SD 52.24, t = 0.29, df = 9.99, $p$ = .77).

### Gaze signal analysis

Fig 4 shows the representative gaze distribution on the monitor in one trial. The gaze of the Expert was distributed over wider areas than the gaze of the Low Skill player. Fig 5 shows the difference in SD of gaze distribution. Two-way ANOVA revealed a significant main effect of skill level (F = 30.99, $p$ < .001, partial $\eta^2$ = .42) and task difficulty (F = 29.23, $p$ < .001, partial $\eta^2$ = .58). However, there was no interaction effect between skill level and task difficulty. Simple main effect analysis found that SD of horizontal gaze distribution during the Easy Task was higher than that during the Moderate Task and the Hard Task, irrespective of skill level ($p$ < .001 for both). No significant difference for SD of vertical gaze distribution between Expert and Low Skill players was found (Fig 5c and 5d).

The proportion of time that the gaze stayed on each AOI (ratio of gaze distribution) is shown in Fig 6. In Area i, there was a significant difference in skill level (Wilcoxon Rank-Sum test, $p$ < .001) while there was no significant difference in task difficulty. In Area ii, there was no significant difference in skill level or task difficulty. In Area iii, a significant difference in skill level was found (Wilcoxon Rank-Sum test, $p$ = .02) while there was no significant difference in task difficulty. In Area iv, there was no significant difference in skill level or task difficulty. In Area v, there was no significant difference in skill level. However, a Kruskal-Wallis test detected a significant difference in task difficulty ($p$ < .001). The time proportion of the Easy Task in Area v was significantly shorter than the time spent on the Moderate Task or the Hard Task (Wilcoxon Rank-Sum test, $p$ < .001 and $p$ = .014, respectively). In Area vi, there was no significant difference in skill level or task difficulty.

**Classification of type of gaze movement.** Fig 7 shows the average percentage of gaze movements (saccade and fixation). For saccade percentage, a significant main effect of skill level was found (F = 8.07, $p$ = .007, partial $\eta^2$ = .15). There was no significant main effect of task difficulty, nor interaction between skill level and task difficulty. These results indicate that the saccade percentage of Expert players was significantly higher than that of Low Skill players, irrespective of task difficulty.

For fixation percentage, a significant main effect of skill level and task difficulty was observed (F = 4.70, $p$ = .03, partial $\eta^2$ = .08 and F = 4.05, $p$ = .02, partial $\eta^2$ = .15, respectively). However, there was no interaction. Simple main effect analysis indicated that the fixation percentage in the Moderate Task was significantly higher than in the Easy Task, irrespective of skill level ($p$ = .02).

**Characteristics of saccade.** Fig 8 shows the average of the saccade velocity (a, b), saccade number (c, d), and saccade length (e, f). The saccade velocity of Expert players was significantly

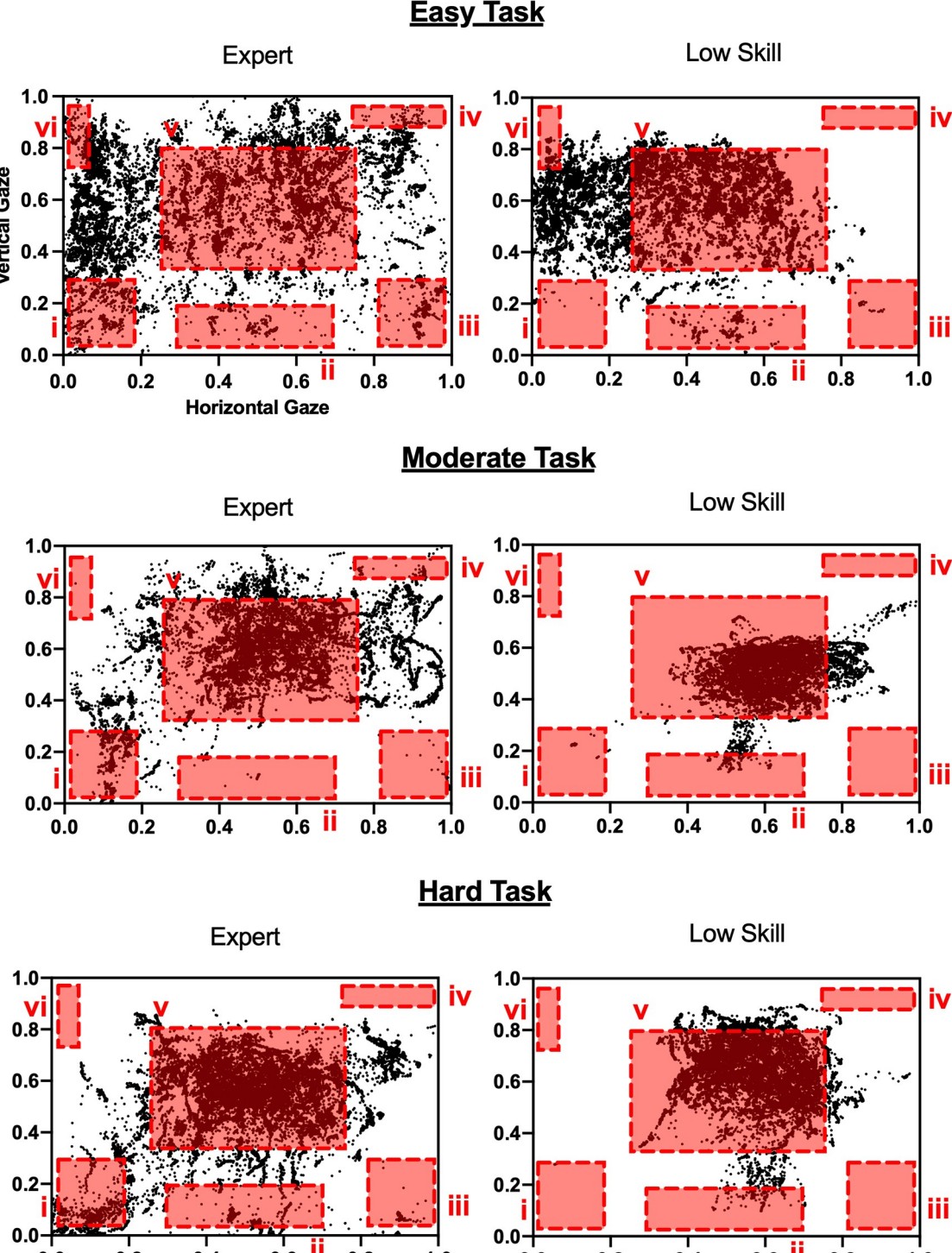

**Fig 4. Examples of gaze distribution.** Each black dot indicates one sample of measured gaze location. The darker areas represent denser gaze location concentrations, meaning that gaze movement was concentrated on that area. Each red square represents an AOI (See Fig 2). Areas which are not included in an AOI have relatively low importance. Vertical axis and horizontal axis are the numerical values obtained by normalizing the monitor size.

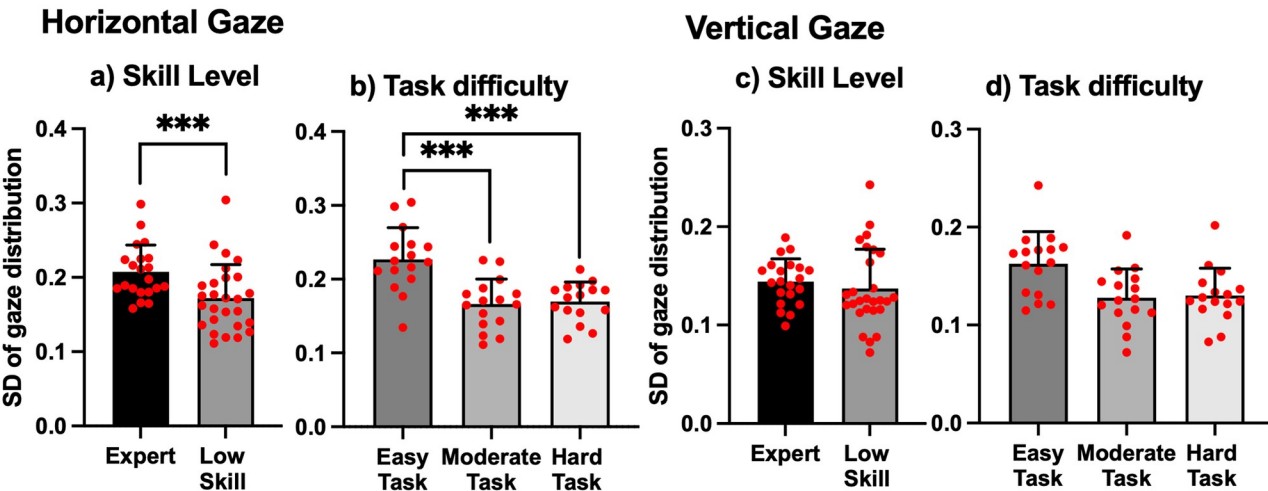

**Fig 5. Gaze distributions.** a, b: SD of gaze distribution in the horizontal direction. c, d: SD of gaze distribution in the vertical direction. Each red circle represents an individual data point. Significance level was set at *** $p < .001$.

faster than that of Low Skill players (Wilcoxon Rank-Sum test, $p = .02$). However, there was no significant difference in task difficulty. For saccade number, a main effect of skill level was found (F = 8.60, $p = .005$, partial $\eta^2 = .15$). However, there was no main effect of task difficulty or interaction. For saccade average length, there was no significant main effect or interaction effect.

## Discussion

The purpose of this study was to examine the gaze control strategy of the StarCraft experts (Expert) during a game which requires task-switching abilities, by comparing Expert with lower skilled players (Low Skill). The Expert players showed significantly higher performance scores than the Low Skill players in the Moderate Task and the Hard Task. Since the number of jobs that participants had to perform simultaneously in the Easy Task was small, the Easy Task was likely so easy that it was unable to reveal group differences, probably due to a ceiling effect. However, it seems that two of the tasks modeled (Moderate Task and Hard Task) successfully discriminated between the abilities of the two subject groups. There was no significant difference in APM score between Expert and Low Skill players. This result indicates that the better performance of the Expert players did not depend upon the number of keys pressed on the keyboard or moves and clicks of the mouse. The superior performance of accomplished StarCraft players in this kind of task-switching could be due to their specific gaze movement. Indeed, some differences were observed between Expert and Low Skill players in gaze control. First, Expert players showed a wider gaze distribution than did the Low Skill players (Fig 5a). This could mean that Expert players scanned a wider area of the screen, which likely helped them to obtain more information. In RTS esports such as StarCraft, different pieces of important information are distributed all over the screen. Thus, when playing such games, it is necessary to rapidly switch attention between multiple points and areas on the screen and maintain this information in working memory. The games utilized in the present study provided many situations in which different procedures had to be carried out in different places in parallel. In such situations, it is important to obtain as much information as possible and respond accurately and quickly during the game. Second, the ratio of saccade was larger and fixation time was shorter in the Expert than in the Low Skill players, suggesting that StarCraft experts

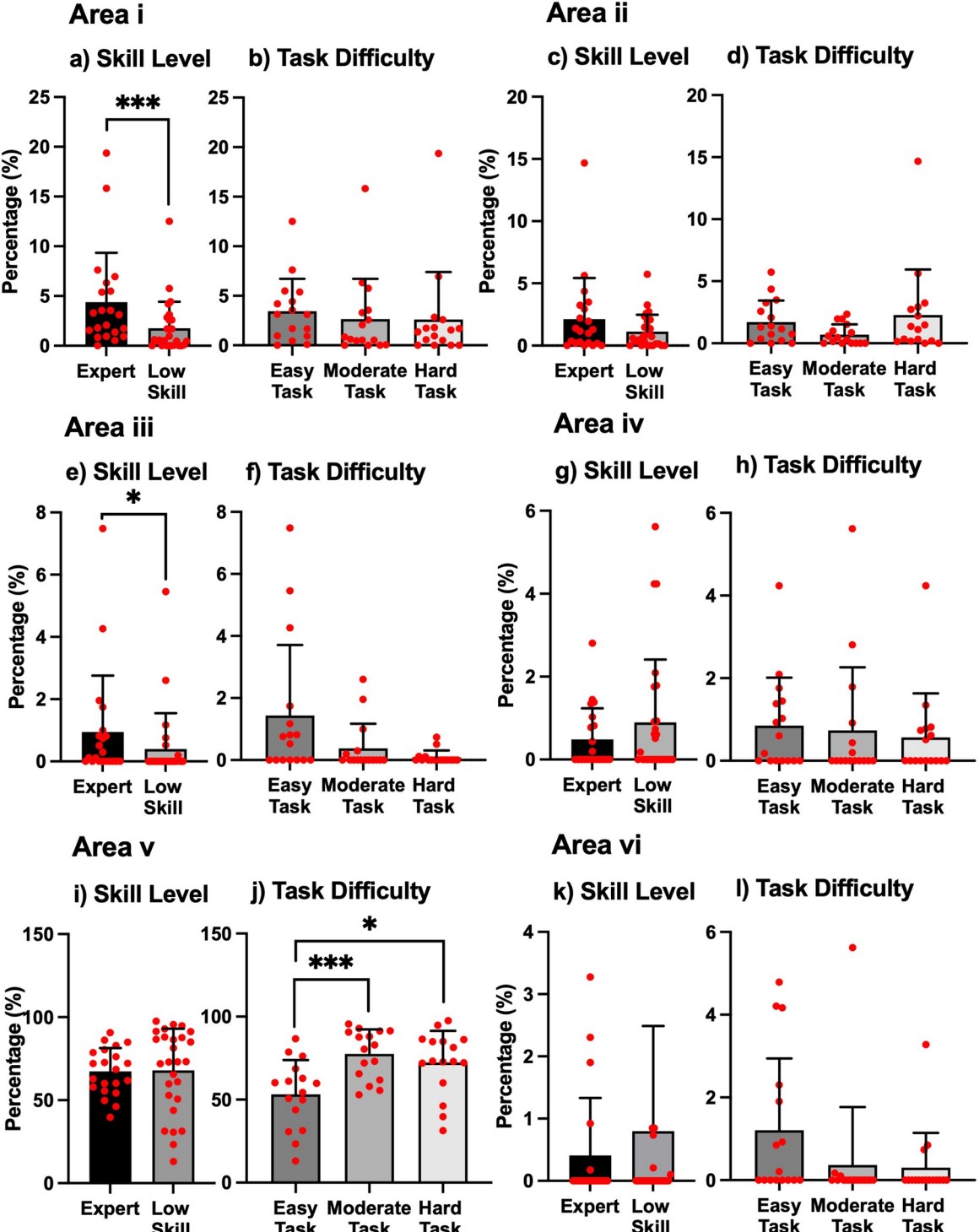

**Fig 6. Ratio of gaze distribution in six areas.** Each red circle represents an individual data point. a: Significance level was set at ***$p <$ .001, e: Significance level was set at *$p <$ .05, j: Significance level was set at * $p <$ .017 and *** $p <$ .001.

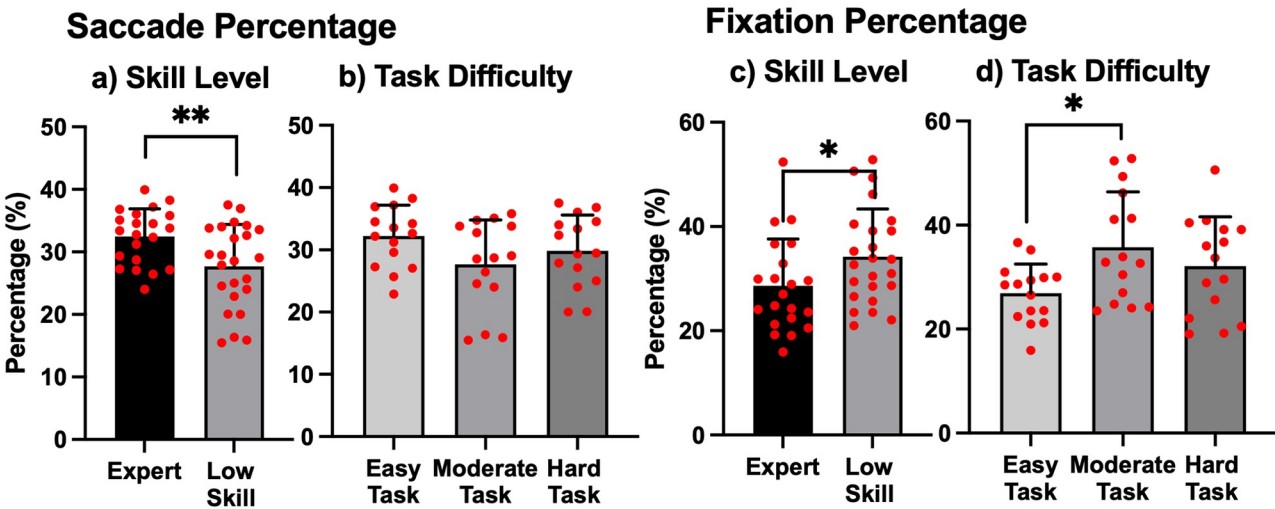

**Fig 7. Classification of gaze movement.** Each data shows the percentage that each gaze movement (saccade and fixation) occupies when performing a task by applying the Eye Movement Detector in Pupil-Player. Each red circle represents an individual data point. The significance level was set at $^{**}p <$ .01 for saccade and $^*p <$ .05 for fixation.

frequently use saccade to quickly change their gaze position and accumulate more information through their shorter fixation time. Third, the results of AOI analysis suggest that Expert players directed a higher proportion of their gaze into Area i and Area iii than the Low Skill players did. Area i contains important information concerning the overall flow of the task; e.g., the information in Area i indicates which Zone should be checked first. This result indicates that the Expert players placed more importance on the overall flow than the Low Skill players did. In order to achieve a high performance level, players must produce normal units; in order to produce normal units, participants must check Area iii. Thus, in order to achieve a high performance level, players have to check Area iii periodically and this could be why Expert players concentrated a higher proportion of their gaze in Area iii. Finally, the higher percentage of fixation in the Low Skill than in the Expert players (Fig 7c and 7d) indicates that the Low Skill players needed longer fixation duration to absorb information and therefore needed to keep their gaze on one location longer.

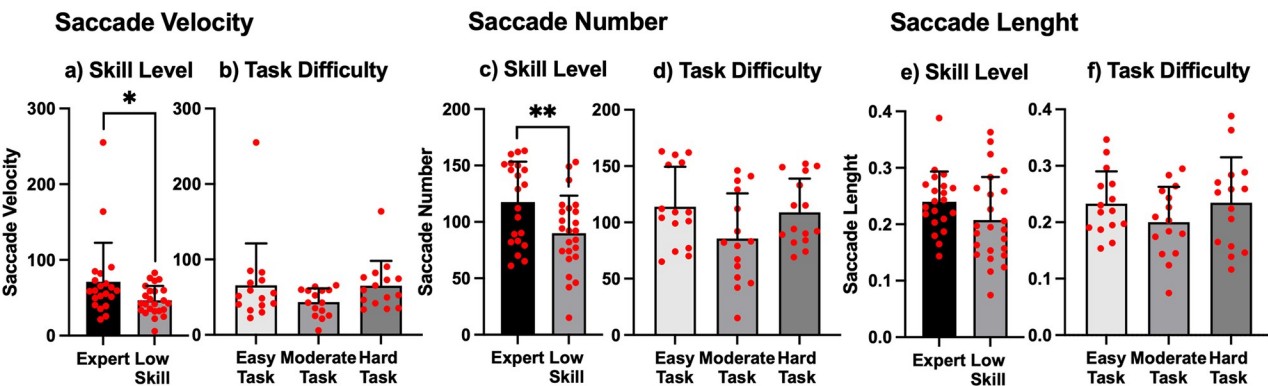

**Fig 8. Saccade characteristics.** a, b: saccade velocity, c, d: saccade number, e, f: saccade length. Each red circle represents an individual data point. The significance level was set at $^*p <$ .05 and $^{**}p <$ .01.

The saccade velocity of Expert players was significantly faster than that of Low Skill players, and the number of saccade movements in the Expert players was significantly greater than in the Low Skill players. Thus, the Expert players shifted their gaze more frequently and more quickly from area to area than the Low Skill players did (for example, from Area i to Area v). These eye movements are advantageous to processing multiple stimuli and successfully playing games when task-switching ability is required.

A previous study failed to detect a significant difference between esports players and non-esports players in a visual attention skill, probably because of the low difficulty of the task utilized for the test [23]. However, the present study showed significant differences in game performance and gaze control strategy between Expert players and Low Skill players in tasks which required higher task-switching ability. That is, experts in RTS StarCraft could perform saccadic eye movements at higher velocity while playing the actual game. Faster saccade makes it possible to quickly change the gaze point, enabling players to process multiple stimuli on the screen more quickly. The present study suggests that StarCraft experts utilized this specific visual ability while playing.

It is well known that skilled athletes in general sports also show different gaze behavior from novice or low skill players. For example, in one-on-one defensive situations in soccer, novice players mostly watch the ball, while experienced players watch not only the ball but also the knee and hip of the opposing player [24], probably because experienced players pay attention more broadly to the overall movement of their opponent. Similarly, in StarCraft players must pay attention to multiple stimuli. The current study may expand the knowledge of gaze control strategy during sports, i.e., the strategy of distributing the gaze to wide areas in the visual field might be common when players in either general sports or RTS esports need to pay attention to multiple information streams simultaneously. In addition, it has been proposed that athletes in general sports use different gaze control strategies depending on the situation [25]. For example, elite basketball players fix their gaze on the hoop significantly longer than novices do in a free throw situation [26]. In this case, throwing the ball into the hoop is the only requirement, so it is advantageous for players to fix their gaze on the hoop. From the above, we can hypothesize that esports players would probably also utilize different gaze strategies depending on the game genre. However, this remains to be elucidated in future studies.

How are StarCraft experts able to exert gaze control that would be specific to them? Perhaps StarCraft experts have better functionality in vision-related brain regions thanks to long-term training in esports proficiency, which requires specific visual functions and task-switching ability. Indeed, a previous study revealed an increase in gray matter volume of the frontal eye field (FEF) in adults older than 55 after two months if training in playing AVG, and the subjects obtained precise and delicate gaze control [27]. Since it has been clarified that saccade and the FEF have a close relationship [28], the plastic changes in the FEF likely result in superior saccade control. In addition to the FEF, an increased specific connection between occipital and parietal areas has been confirmed in expert RTS game players compared to non-RTS game players [29]. Therefore, StarCraft experts might also have outstanding function in the FEF and an increase in the interhemispheric connections of the visual cortex.

Finally, what kind of training can improve the performance level of lower skill StarCraft players? We recommend gaze movement training which includes longer saccadic gaze movements that cover a wider area of the screen. This training method is likely to improve the gaze control ability of StarCraft players with lower skill or novice players. In physical sports, such as soccer and baseball, training players to follow a moving object with their eyes is known to be effective. In particular the NeuroTracker system, which tracks multiple 3D objects by using gaze direction, can improve task-switching ability and the visual attention of athletes [30, 31]. Thus, it is possible that gaze movement training will also contribute to improving the

performance level of the StarCraft player with lower skill. Additionally, Dale and Green discovered that training using RTS games improves task-switching ability [12]. This suggests that our findings (StarCraft experts have a specific gaze control strategy) can be generalized to other esports games, and we believe that the specific gaze control strategy used by expert esports players is the source of their superior performance. Thus, knowledge about the gaze control strategy used by successful esports players would be helpful for esports coaching and for devising new training methods.

## Limitations

In this study, we did not record the participants' experience in playing other esports or their years of esports experience. Therefore, we cannot rule out the possibility that a history of playing other esports affected the difference in gaze movement between the Expert players and the Low Skill players. Furthermore, we were not able to strictly control participants' head movements, and the possibility that the difference in gaze movement was caused by head movement cannot be excluded. Additionally, the tasks were performed in the same order, so we cannot completely eliminate the order effect. However, the purpose of the current study was not to ask the effect of task difficulty but to ask whether there is a difference in performance and gaze movements between esports experts and lower skilled players. For this reason, the conclusion of the current study should not be affected by order effect.

## Conclusion

The present study, using StarCraft as a model of esports, suggests that expert StarCraft players show wider gaze movement, covering all areas of the screen, and especially pay more attention to the overall flow of the game (Area i) compared to the players with lower skills. This wider gaze movement is actualized by their faster and longer saccade than the saccade exhibited by the players with lower skills. These specific gaze control strategies in experts are likely related to the higher t require task-switching ability.

## Supporting information

**S1 File. Detail of easy task.**
(DOCX)

**S2 File. Detail of moderate task.**
(DOCX)

**S3 File. Detail of hard task.**
(DOCX)

**S4 File. Detail of operation method.**
(DOCX)

**S5 File. Total gaze movement data of participant.**
(DOCX)

**S6 File. Detail of G$^*$ power protocol.**
(DOCX)

## Acknowledgments

Authors thank Dr. Candace O'Connor for the English editing.

## Author Contributions

**Conceptualization:** Inhyeok Jeong, Kento Nakagawa, Kazuyuki Kanosue.

**Data curation:** Inhyeok Jeong.

**Formal analysis:** Inhyeok Jeong.

**Investigation:** Inhyeok Jeong.

**Methodology:** Inhyeok Jeong.

**Project administration:** Kento Nakagawa, Kazuyuki Kanosue.

**Resources:** Rieko Osu, Kazuyuki Kanosue.

**Supervision:** Kento Nakagawa, Rieko Osu.

**Validation:** Inhyeok Jeong.

**Visualization:** Inhyeok Jeong.

**Writing – original draft:** Inhyeok Jeong.

**Writing – review & editing:** Inhyeok Jeong, Kento Nakagawa, Rieko Osu, Kazuyuki Kanosue.

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
