## [Decision Letter · Decision Letter 0]

18 May 2021

PONE-D-21-09912

Gaze control ability in e-sports experts

PLOS ONE

Dear Dr. Nakagawa,

Thank you for submitting your manuscript to PLOS ONE. After careful consideration, we feel that it has merit but does not fully meet PLOS ONE’s publication criteria as it currently stands. Therefore, we invite you to submit a revised version of the manuscript that addresses the points raised during the review process.

We look forward to receiving your revised manuscript.

Kind regards,

Greg Wood, PhD

Academic Editor

PLOS ONE

Journal Requirements:

3. We note you have included a table to which you do not refer in the text of your manuscript. Please ensure that you refer to Table 2 in your text; if accepted, production will need this reference to link the reader to the Table.

4. We note that Figures 1, 2. S1, S2, S3 in your submission contain copyrighted images. All PLOS content is published under the Creative Commons Attribution License (CC BY 4.0), which means that the manuscript, images, and Supporting Information files will be freely available online, and any third party is permitted to access, download, copy, distribute, and use these materials in any way, even commercially, with proper attribution. For more information, see our copyright guidelines: http://journals.plos.org/plosone/s/licenses-and-copyright.

4.1.    You may seek permission from the original copyright holder of Figures 1, 2. S1, S2, S3 to publish the content specifically under the CC BY 4.0 license.

4.2.    If you are unable to obtain permission from the original copyright holder to publish these figures under the CC BY 4.0 license or if the copyright holder’s requirements are incompatible with the CC BY 4.0 license, please either i) remove the figure or ii) supply a replacement figure that complies with the CC BY 4.0 license. Please check copyright information on all replacement figures and update the figure caption with source information. If applicable, please specify in the figure caption text when a figure is similar but not identical to the original image and is therefore for illustrative purposes only.

Reviewers' comments:

Reviewer's Responses to Questions

**Comments to the Author**

1. Is the manuscript technically sound, and do the data support the conclusions?

Reviewer #1: Yes

Reviewer #2: Partly

2. Has the statistical analysis been performed appropriately and rigorously? 

Reviewer #1: Yes

Reviewer #2: Yes

3. Have the authors made all data underlying the findings in their manuscript fully available?

Reviewer #1: No

Reviewer #2: Yes

4. Is the manuscript presented in an intelligible fashion and written in standard English?

Reviewer #1: No

Reviewer #2: Yes

5. Review Comments to the Author

Reviewer #1: REVIEW

Gaze control ability in e-sports experts

Jeong Inhyeok, Kento Nakagawa, Rieko Osu, Kazuyuki Kanosue

VERDICT: Major Revisions

Recommendations:

The authors present a study that examines the behavioural and gaze differences between novice and expert StarCraft players. The research question is of interest to the field, the methodology and statistical analyses are generally sound and the findings are interesting. However, there are major revisions that need to be addressed in the introduction and justification of the study, the reporting of results and the discussion of the findings that would need to be addressed before the manuscript could be considered for publication. Also, the grammer is very poor despite the authors report that it had been examined for proper use of the English language. In many instances, wording is awkward and the plural of a word is required and not used among other grammatical issues. This significantly takes away from the readability of the paper and presentation of the work and a thorough edit of the manuscript’s grammer by someone whose first language is English is required.

**All Page and Line number indicators below are formatted as P#L###-###.

INTRODUCTION:

P2L35: Please change ‘e-sports’ to ‘esports’ throughout manuscript, unless starting a sentence, which in that case, ‘Esports’ is correct.

P3L39: esports is not a subcategory of AVG, this would imply that all esports are AVGs. Instead, AVGs, when played competitively and/or professionally, can be considered esports. Please reword.

P3L40-41: It isn’t the expanding market that has led to research on cognition in esports. I would reword this to highlight that esports players are often considered cognitive athletes and it’s this foundation, in conjunction with the increased popularity of esports that has led to an increase in research attention toward the cognitive benefits associated with esports and VG more broadly. See and reference Campbell et al., 2018.

General Intro: Missing a key reference regarding superior info processing and task-switching abilities of AVG’s compared to Non Gamers that would support the authors introductory arguments. See Kowal et al., 2018.

P3L49: AVG should be AVGs (plural). Also to this line…what is the difference between cognitive and visual function? Visual function as in a change in the function of the retinal cells? The visual cortex? Or do you mean visual perception here? Would be careful with wording here as if it’s a change in saccade pattern for example, this isn’t a visual function change but an adaptation to a change in cognitive function or strategy.

P4L60-66: Why is looking at in-game saccade behaviour important? What evidence is there from other cognitive aspects that results in clinical testing and in game testing might result in different conclusions being made? This is a key point the authors must address in their justification of this study.

P4L67: Starcraft is a real time strategy game (RTS). As it is not considered an AVG, previous arguments around AVGs and enhanced cognition or saccade performance do not justify examining Starcraft here. The authors would need to rework their entire introduction to justify examining gaze patterns for this style of video game, as opposed to basing all of their evidence around AVGs.

P4L70-71: Current hypothesis is not an actual hypothesis. Given the purpose is to examine gaze control strategy between novices and expert SC players, the hypothesis should explicitly state who will be expected to show superior gaze control strategy and what metric will indicate this. Please edit hypothesis.

METHODS:

N=16, N Experts = 7, N Novices = 9

The N's seem very low. Can the authors provide a power analysis justifying the minimum number of participants required to be able to meaningfully detect the effects they hope to detect?

P5L82: Authors indicate that all participants have experience playing Starcraft. This being the case, it is incorrect to call any of the participants ‘Novices’ and instead, the groups should be separated based on skill level (Expert or high skill vs. Low Skill).

Did the authors control for other video games that individuals played? Other genres? How often they currently spend playing video games (i.e., hours per week). These are all important considerations in defining the groups that should be considered and added where appropriate. Otherwise, these aspects should be included within the limitations section of the discussion.

P10L168-172: APM does not measure speed of hand movement per se. Hands would have to be kinematically tracked using 3d Motion capture hardware and software. Please revise wording here.

P11L178: I would like to see the distribution of monitor sizes used by participants in the study reported. It’s very easy to say there is no difference but what test was used? Were the monitor sizes across participants normally distributed. I would guess not. How did the different metrics pay out for different monitor sizes?

P11L179-180: How was head position maintained at 40 inches? How was this ensured or tracked? What was the variability in head position across 3 min trials? These need to be explained as gaze performance conclusions can be confounded by this.

P13L222: Were data tested for normality and homogeneity of variance? Please provide these details.

RESULTS:

P14L236: How are data presented: Means +- SD or SE or 95%CI??? Please state.

P15L265: Incorrect to say ‘Simple main effect analysis found that the AOI of the Easy Task was significantly larger than that of the Hard Task (p = .02)’ Instead, Proportion of time gaze stayed in area 3 was greater for easy vs hard task. Please revise wording around this finding.

P16L274: State what the finding was? Who had the higher proportion of saccades?

DISCUSSION:

P17L299: This was not your purpose. Your purpose was to examine and compare the gaze control strategy between novices and expert SC players. Please rephrase.

P17L300-301: The authors state ‘The experts showed significantly higher performance scores than the novices in all three Tasks, as expected. Therefore, it seems that the three Tasks modeled successfully discriminated between the abilities of the two subject groups’. This was not the case. Skill level differences could not be found in the easy task. Why might experts and novices differ only for moderate and hard tasks. Discussion should be provided around this.

P18L312-313: I would like the authors to speak to their interpretation that ‘the ratio of saccade was larger in the experts than in the novices, suggesting that e-sports experts frequently use saccade for quickly obtaining the information they need.’ This seems to contradict previous literature that suggests information is not gathered during saccades but during fixations. They seem to note this when discussing their fixation ratio findings. I would like the authors to contrast this with expertise literature on quiet eye etc., which would suggest that longer final fixations durations are a marker of expertise.

P18L326: Starcraft is a RTS, not an AVG. Please revise here and throughout.

P19L331-333: This argument that experts direct focus to the foot in soccer does not support the authors claim that displaying more saccades across a larger horizontal field of view is a marker of expertise. In fact, it suggests the opposite, that experts in traditional sports maintain focus on a small area whereas experts in esports do not. I hope they might be able to clarify and add to their argument here.

FIGURES:

Figure 4: incorrect reporting of statistical findings. Firstly, graphs should be shown as two plots (one comparing Novice to expert with data pooled for difficulty in the two bars – then a significance star between the two bars to show the main effect of skill level. Then the second plot, 3 bars with expertise pooled in each of the easy, moderate and hard bars, with star over easy to show it was different from the other two bars. The vertical graph can be separated into two plots in the same way, such that you have figures 4a, 4b, 4c, and 4d with a and b plots showing horizontal data and c and d plots showing vertical data.

Figures 5, 6 and 7 can all be adapted in the same way. If the authors feel strongly about presenting the individual skill level and expertise bars, I would recommend the presentation of the data in this way be done in the form of supplementary material figures. The figures as they are, are not effective in their display of the main findings and support for or against the purpose and hypotheses laid out in this manuscript.

Reviewer #2: Comments to the authors

General Comments

Thank you for your manuscript that investigates the difference in aspects of gaze control between experts esports players and novices while playing the game StarCraft. However, I will address major concerns that could encourage authors to improve the manuscript.

First of all, the introduction section needs some rework, starting from implementing or acknowledging the current standardise definitions and conceptualisation of esports. This, as authors are using a wrong spelling of the word esports and it is important to clarify that esports is not a subcategory of action video games (AVG) (see review, (Pedraza-Ramirez et al., 2020). Even though authors are acknowledging the current literature of video games and cognitive functions, it is recommended to consider including not only the single studies in the video gaming literature but as well the recent meta-analysis that have found mixed evidence on the effects of video game practice on cognition (e.g., Bediou et al., 2018; Sala et al., 2017) and the current state of the art and relevant studies in esports (e.g., Li et al., 2020; Pedraza-Ramirez et al., 2020; Thompson et al., 2019; Toth et al., 2019). Therefore, these are important characteristics that required attention from the gaming research to the field of esports.

In general, the manuscript needs to improve at the theoretical level. This, because it was not mentioned any theoretical background that can help to support the aim of the study and develop clear assumptions. Authors identified different areas of investigation in relation to the aim of the study such as cognitive functions, multitasking or motor-cognitive processes, however, it is not clear the theoretical foundation. However, due to the lack of specificity the manuscript is not doing a good job showing how this research will contribute to the extension of knowledge. Even though it was shown in the manuscript a great number of studies supporting the importance to study visual functions and saccade control ability in video game players, authors need to consider, first, a theoretical account that will help to support the hypothesis. Second, acknowledging the considerations towards the issues of lack of rigorous methodological designs in video game research and esports (e.g., not grouping video game genres) (see Dale & Green, 2017). Third, differentiating and bring specificity whether the study is on esports or video games. Consequently, the current state of the hypothesis is too vague and do not have a supporting evidence, thus, affecting the trustworthiness of the study.

Additionally, the method section needs rework, as the procedure is not clearly explained step by step for replicability and is missing important information. For example, we do not know whether was controlled that participants were engaging in other esports or video games or if it was asked for the years of experience playing StartCraft or other games. This, as experience and the type of game played may be crucial elements when investigating expertise differences in relation to cognitive abilities (see Li et al., 2020). Also, authors are not showing an a priori G* Power calculation to estimate the sample size required. Thus, clarifying and addressing the reliability of the methods used, clarity in the participants' background, and the procedure of the data collection brings rigour and reliability of the methodological design. Although the experimental task is very innovative and can add methodological value to research and its applicability in esports due to its ecological validity, I am wondering whether the test is supported by a previous investigation or if it was previously piloted, and if so, it would be required to address the findings so that there are not risks of construct validity. Lastly, I think that they behavioural data could be interesting form a motor learning and motor control perspective, however, I am wondering how this data aligns with the aim of the study. Therefore, integrating the results from this performance variable into the characteristics of gaze control could bring a better understanding of expertise in this manuscript.

Finally, it is necessary to strengthen and highlight the added values and extension of knowledge at the introduction and support it in the discussion. The discussion section will benefit of integrating supporting evidence align with the findings. I think authors are doing a good job in lines 324 to 339 by discussing the results with previous evidence and expanding the knowledge.

The comments are mean to encourage authors to improve the quality and specificity of the direction of the manuscript.

Suggested Reference:

Bediou, B., Adams, D. M., Mayer, R. E., Tipton, E., Green, C. S., & Bavelier, D. (2018). Meta-analysis of action video game impact on perceptual, attentional, and cognitive skills. Psychological Bulletin, 144(1), 77–110. https://doi.org/10.1037/bul0000130

Dale, G., & Green, C. S. (2017). The Changing Face of Video Games and Video Gamers: Future Directions in the Scientific Study of Video Game Play and Cognitive Performance. Journal of Cognitive Enhancement, 1(3), 280–294. https://doi.org/10.1007/s41465-017-0015-6

Li, X., Huang, L., Li, B., Wang, H., & Han, C. (2020). Time for a true display of skill: Top players in League of Legends have better executive control. Acta Psychologica, 204. https://doi.org/10.1016/j.actpsy.2020.103007

Pedraza-Ramirez, I., Musculus, L., Raab, M., & Laborde, S. (2020). Setting the scientific stage for esports psychology: a systematic review. International Review of Sport and Exercise Psychology, 0(0), 1–34. https://doi.org/10.1080/1750984X.2020.1723122

Sala, G., Tatlidil, K. S., & Gobet, F. (2017). Video game training does not enhance cognitive ability: A comprehensive meta-analytic investigation. Psychological Bulletin, 144(2), 111–139. https://doi.org/10.1037/bul0000139

Thompson, J. J., Mccoleman, C. M., Blair, M. R., & Henrey, A. J. (2019). Classic motor chunking theory fails to account for behavioural diversity and speed in a complex naturalistic task. PLoS ONE, 14(6), 1–24. https://doi.org/10.1371/journal.pone.0218251

Toth, A. J., Kowal, M., & Campbell, M. J. (2019). The Color-Word Stroop Task Does Not Differentiate Cognitive Inhibition Ability Among Esports Gamers of Varying Expertise. Frontiers in Psychology, 10(December). https://doi.org/10.3389/fpsyg.2019.02852

6. PLOS authors have the option to publish the peer review history of their article (what does this mean?). If published, this will include your full peer review and any attached files.

Reviewer #1: No

Reviewer #2: **Yes: **Ismael Pedraza-Ramirez

---

## [Author Response · Author response to Decision Letter 0]

19 Aug 2021

Response to Editor and Reviewers

Dear Academic Editor and Reviewers,

Thank you very much for your feedback, comments and suggestions. Our responses to your comments are as follows:

Editor:

Comment 1: Please ensure that your manuscript meets PLOS ONE's style requirements, including those for file naming.

Response: The style of the manuscript has been corrected. To be specific, the name of the figures file have been changed to follow PLOS ONE requirements. 

Comment 2: Please provide additional details regarding participant consent.

Response: We have corrected as follows:

Change: [P5L88-90] Before the experiment, we verbally provided information about the contents and concepts of this research along with the instruction documents. After that, we obtained verbal informed consent from all subjects.

Comment 3: We note you have included a table to which you do not refer in the text of your manuscript. Please ensure that you refer to Table 2 in your text.

Response: We have referred to Table 2 as follows:

Change: [P10L165] The above mentioned conditions for game over are summarized in Table 2.

Comment 4, 4.1, 4.2: We note that Figures 1, 2. S1, S2, S3 in your submission contain copyrighted images.

Response: Thank you for your consideration of copyrighted images. The website of the copyright holder (Blizzard Entertainment), clearly states that images can be freely used regardless of copyright for educational purposes (see https://www.blizzard.com/en-us/legal/dd76b654-f2c4-4aaa-ba49-ca3122de2376/blizzard-video-policy). Additionally, we received a response from the copyright team of the copyright holder via email saying that our use of their material in Figures 1, 2 and S1, S2, S3 are acceptably used for academic purposes. To be specific, the copyright holder has clearly said that all StarCraft images can be used in journal publications.

Reviewer #1:

Recommendations: The authors present a study that examines the behavioral and gaze differences between novice and expert StarCraft players. The research question is of interest to the field, the methodology and statistical analyses are generally sound and the findings are interesting. However, there are major revisions that need to be addressed in the introduction and justification of the study, the reporting of results and the discussion of the findings that would need to be addressed before the manuscript could be considered for publication.

Response: Thank you for your recommendations. The introduction and the reporting of results and　discussion have been reworked. 

Also, the grammar is very poor despite the authors report that it had been examined for proper use of the English language. In many instances, wording is awkward and the plural of a word is required and not used among other grammatical issues. This significantly takes away from the readability of the paper and presentation of the work and a thorough edit of the manuscript’s grammer by someone whose first language is English is required.

Response: Thank you for your suggestion. According to your suggestion, the manuscript has been　proofread by a professional proofreading service again.

INTRODUCTIONS:

P2L35: Please change ‘e-sports’ to ‘esports’ throughout manuscript, unless starting a sentence, which in that case, ‘Esports’ is correct.

Response: We are sorry for using incorrect words. The word “e-sports” has been replaced with “esports”　and “Esports”

P3L39: esports is not a subcategory of AVG, this would imply that all esports are AVGs. Instead, AVGs, when played competitively and/or professionally, can be considered esports. Please reword.

Response: Thank you for your suggestion. Definition of esports has been reworded and a reference has been added as follows:

Change: [P2L35-36] Irrespective of game genre, when a game is played competitively and professionally, it can be considered esports [2].

P3L40-41: It isn’t the expanding market that has led to research on cognition in esports. I would reword this to highlight that esports players are often considered cognitive athletes and it’s this foundation , in conjunction with the increased popularity of esports that has led to an increase in research attention toward the cognitive benefits associated with esports and VG more broadly. See and reference Campbell et al., 2018.

Response: We appreciate the suggestion. The reason why research on esports has been expanded has been added.

Change: [P3L39-40] In addition, the increasing popularity of esports has resulted in further research on the cognitive benefits associated with esports and video games [3]

General Intro : Missing a key reference regarding superior info processing and task-switching abilities of AVG’s compared to Non Gamers that would support the authors introductory arguments.

Response: Thank you for your suggestion. According to your suggestion, a key reference has been added 

as follows:

Change: [P3L40-41] Kowal et al. [16] found that the task-switching ability of action video game (AVG) players is higher than that of non-AVG players.

P3L49: AVG should be AVGs (plural). Also to this line…what is the difference between cognitive and visual function? Visual function as in a change in the function of the retinal cells? The visual cortex? Or do you mean visual perception here? Would be careful with wording here as if it’s a change in saccade pattern for example, this isn’t a visual function change but an adaptation to a change in cognitive function or strategy.

Response: Thank you for your suggestion. AVG has been revised to AVGs. We are sorry for using　ambiguous words (cognitive and visual function). We removed the ambiguous “visual function”　in the whole paragraph. Additionally, the definition of “cognitive” has been added.

In the current study, cognitive function refers to the mental process involved in obtaining information and knowledge.

Change: [P3L43-44] In addition, a recent meta-analysis also found that video game practice has positive effects on cognitive function [14] (the mental processing required to obtain information and knowledge [6]).

P4L60-66: Why is looking at in-game saccade behaviour important? What evidence is there from other cognitive aspects that results in clinical testing and in game testing might result in different conclusions being made? This is a key point the authors must address in their justification of this study.

Response: Thank you for your suggestion. In esports, especially in real-time strategy (RTS) games, multiple stimuli appear simultaneously and players are required to judge how to respond to those stimuli. For high RTS performance, players need to check multiple information sources, switching from one to the other as fast as possible. In this situation, players should make fast saccadic gaze movements. In order to test this hypothesis, ordinary clinical testing using a simple task would not be appropriate because simple tasks present less stimuli and do not require the degree of multitasking required in RTS. Indeed, Murphy and Spencer (2009) argue that there was no significant difference between esports players and non-esports players in visual attention skill, which was verified only in a simple attention blink task used in the laboratory

. The possible reason why this previous research did not find better visual ability in esports players would be that the simple laboratory task may be too easy because multitasking was not incorporated in the task. For this reason, in the current study we set the task to be like a real game in order to detect the strategy esports players use to control their gaze during game playing. 

Information about the importance of measuring saccade behavior and the difference between a clinical test and a gaming test based on the evidence has been added as follows:

Change: [P3L51-P4L67] Indeed, in RTS, multiple information streams appear simultaneously in different locations on the monitor screen, and players must absorb and select information of importance to them. Based on the information obtained, they must judge how to make an appropriately timed response to the stimuli. Thus, multitasking ability could be the key element underlying higher performance in RTS. RTS players must move their gaze over multiple areas on the monitor to win the game. Therefore, esports players would use more ballistic gaze movements, called saccade, to enable them to process multiple stimuli more quickly. However, the characteristics of gaze strategy in esports players are still unknown. Therefore, the purpose of the current study was to examine the gaze control of RTS expert players. We hypothesized that RTS experts in esports would show superior gaze control strategy (wide and fast saccadic gaze movement) which is evaluated by measuring gaze distribution and saccadic movements. Clarifying the gaze control strategy used by successful esports players would be beneficial for esports coaching, and also for expanding esports markets. 

Murphy and Spencer [17] argue that there was no significant difference between esports players and non-esports players in visual attention skill; this conclusion was verified by using only a simple attention blink task in a laboratory, which was unrealistic because there was no multitasking involved. Therefore, in the current study for analyzing how esports players control their gaze while playing a game, we set them a task which was similar to what occurs in real RTS games; we required players to exhibit higher multitasking abilities, and to cope with increasing task difficulty. 

P4L67: Starcraft is a real time strategy game (RTS). As it is not considered an AVG, previous arguments around AVGs and enhanced cognition or saccade performance do not justify examining Starcraft here. The authors would need to rework their entire introduction to justify examining gaze patterns for this style of video game, as opposed to basing all of their evidence around AVGs.

Response: We appreciate your comments. We changed the genre of StarCraft from AVG to RTS. Additionally, information about cognitive functions in RTS players and the reason why it is important to focus on gaze patterns has been added as follows:

Change: [P3L45-50] The real-time strategy (RTS) game is another kind of video game. In RTS games players are required to multi-tasking using complex strategies. AVG and RTS have similar positive effects on cognitive processes (both AVG and RTS improve reaction time and problem-solving ability) [10]. Meanwhile, Dobrowolski et al. [4] found that RTS players have superior cognitive function compared to that of AVG players. Indeed, RTS game training improves task switching ability [18]. In order to switch between tasks, it is necessary to accurately perceive stimuli presented on the monitor screen through proper gaze control. 

P4L70-71: Current hypothesis is not an actual hypothesis. Given the purpose is to examine gaze control strategy between novices and expert SC players, the hypothesis should explicitly state who will be expected to show superior gaze control strategy and what metric will indicate this. Please edit hypothesis.

Response: We agree with your suggestion. We have edited the hypothesis as follows:

Change: [P4L58-59] We hypothesized that RTS experts in esports would show superior gaze control strategy (wide and fast saccadic gaze movement) which is evaluated by measuring gaze distribution and saccadic movements.

METHODS:

N=16, N Experts = 7, N Novices = 9

The N's seem very low. Can the authors provide a power analysis justifying the minimum number of participants required to be able to meaningfully detect the effects they hope to detect?

Response: Thank you for this suggestion. Information about power analysis has been added.

Change: [P5L78-84] Before starting the experiment, we performed a power analysis to estimate the required sample size (RStudio version 1.3). The sample size was calculated using preliminary experimental data related to the distribution of gaze movements of esports experts (Expert) and players with lower skills (Low Skill) (effect size: 0.51; α level: 0.05; power (1-β error probability): 0.80). As a result, the calculated necessary number of participants was seven in each group. According to the result of this power analysis, sixteen participants (15 male, 1 female; 7 Expert, 9 Low Skill; mean age, 22.4 years; age range, 18-28 years) with experience playing StarCraft participated in the present study.

P5L82: Authors indicate that all participants have experience playing Starcraft. This being the case, it is incorrect to call any of the participants ‘Novices’ and instead, the groups should be separated based on skill level (Expert or high skill vs. Low Skill).

Response: Thank you for your consideration. All ‘Novice’ have been changed to ‘Low Skill’. Along with this change, “expert” was also changed to “Expert”. 

Did the authors control for other video games that individuals played? Other genres? How often they currently spend playing video games (i.e., hours per week). These are all important considerations in defining the groups that should be considered and added where appropriate. Otherwise, these aspects should be included within the limitations section of the discussion.

Response: Unfortunately, we did not control for other video games in other genres and collect the participants’ history of esports play. Lack of information of other esports play history could affect the participants’ properties. Additionally, we could not rule out the possibility that other esports play history has affected the difference between the Expert and the Low Skill. A statement about playing history has been added to the limitation section.

Change: [P22L394-396] we did not record the participants’ experience in playing other esports or their years of esports experience. Therefore, we cannot rule out the possibility that a history of playing other esports affected the difference in gaze movement between the Expert and the Low Skill players.

P10L168-172: APM does not measure speed of hand movement per se. Hands would have to be kinematically tracked using 3d Motion capture hardware and software. Please revise wording here.

Response: We strongly agree with your opinion. Thus, “speed of hand” has been changed to “number of clicks”.

Change: [P11L179-180] In order to compare the number of clicks between the Expert and Low Skill players,

P11L178: I would like to see the distribution of monitor sizes used by participants in the study reported. It’s very easy to say there is no difference but what test was used? Were the monitor sizes across participants normally distributed. I would guess not. How did the different metrics pay out for different monitor sizes?

Response: Thank you for your question. Data about monitor size followed homogeneity but not normality. For this reason, Wilcoxon Rank-Sum test was used to check significant difference in monitor size between Expert and Low Skill. No significant difference in monitor sizes between the groups was found (p = .30). Information of monitor sizes (means and SD) has been added.

Change: [P11L186-188] There was no significant difference in the selected monitor size between Expert players (average of monitor size: 27.00 inch ± SD 3.87) and Low Skill players(average of monitor size : 28.77 inch ± SD 4.35) (Wilcoxon Rank-Sum test, p = .30).

P11L179-180: How was head position maintained at 40 inches? How was this ensured or tracked? What was the variability in head position across 3 min trials? These need to be explained as gaze performance conclusions can be confounded by this.

P11L179-180: What was the variability in head position across 3 min trials? These need to be explained as gaze performance conclusions can be confounded by this.

Response: We confirmed the initial distance between each participant and the monitor was 40 inches before starting the task. Since we instructed participants to keep the current position of their heads while sitting on the chair, the head position was unlikely to move very much. However, as we did not track the actual head position, we cannot strictly control the movement of the head. This information has been added to the limitation section.

Change: [P11L188-189] Before the task started, we measured the distance (40 inches) between each participant’s head and their monitor. After measuring the distance, we instructed participants to keep the position of their head steady.

Change: [P22L396-398] Furthermore, we were not able to strictly control participants’ head movements, and the possibility that the difference in gaze movement was caused by head movement cannot be excluded.

P13L222: Were data tested for normality and homogeneity of variance? Please provide these details.

Response: Thank you for your suggestion. As we did not check the normality nor homogeneity of variance in the previous version, we have analyzed them by Levene’s test and Shapior-Wilk’s test in this revision. We found that SD of gaze distribution (vertical gaze), AOI, and saccade characteristic (saccade velocity) did not follow normality and homogeneity of variance. Therefore, these data were reanalyzed by a non-parametric method (Kruskal Wallis test and Wilcoxon rank sum test with Bonferroni’s correction). We have added information about normality and homogeneity of variance as follows:

Change: [P14L234-240] Homogeneity of variance was tested with Levene’s test and normality of variance was tested with Shapior-Wilk’s test. As a result, SD of gaze distribution (vertical gaze), ratio of fixation in each area (AOI), and saccade velocity did not follow a normal distribution of variance and show homoscedasticity. For this reason, these data were calculated by a non-parametric method. When the Kruskal-Wallis test detected significance, the Wilcoxon Rank-Sum test was performed to find the difference between each group of Task difficulties (Bonferroni’s correction (p < .05 divided by 3 tests: significance threshold at .017)). 

[P18L310-312] The saccade velocity of Expert players was significantly faster than that of Low Skill players (Wilcoxon Rank-Sum test, p = .02).

RESULTS:

P14L236: How are data presented: Means +- SD or SE or 95%CI??? Please state.

Response: We are sorry for omitting the statement. In the behavioral data section, data were presented as means ± SD. 

P15L265: Incorrect to say ‘Simple main effect analysis found that the AOI of the Easy Task was significantly larger than that of the Hard Task (p = .02)’ Instead, Proportion of time gaze stayed in area 3 was greater for easy vs hard task. Please revise wording around this finding.

Response: Thank you for your suggestion. In this paragraph (Area of Interest), all data have been recalculated with non-parametric analysis. Thus we have reworked all statistics descriptions.

Change: [P16L281-P17L289] The proportion of time that the gaze stayed on each AOI (ratio of gaze distribution) is shown in Figure 5. In Area i, there was a significant difference of skill level (Wilcoxon Rank-Sum test, p < .001) while there was no significant difference between Task difficulty. In Area ii, there was no significant difference in skill level or Task difficulty. In Area iii, a significant difference of skill level was found (Wilcoxon Rank-Sum test, p = .02) while there was no significant difference in Task difficulty. In Area iv, there was no significant difference in skill level or Task difficulty. In Area v, there was no significant difference in skill level. However, a Kruskal-Wallis test detected significance in Task difficulty (p < .001). The time proportion of the Easy Task was significantly shorter than the time spent on the Moderate Task or the Hard Task (Wilcoxon Rank-Sum test, p < .001 and p = .014, respectively). In Area vi, there was no significant difference in skill level or Task difficulty.

P16L274: State what the finding was? Who had the higher proportion of saccades?

Response: Sorry for the lack of description. We have added the finding briefly as follows: 

Change: [P17L295-299] For saccade percentage, a significant main effect of skill level was found (F = 8.07, p = .007, partial η2 = .15). There was no significant main effect of Task difficulty, nor interaction between skill level and Task difficulty. These results indicate that the saccade percentage of Expert players was significantly higher than that of Low Skill players, irrespective of Task difficulty. 

DISCUSSION:

P1L299: This was not your purpose. Your purpose was to examine and compare the gaze control strategy between novices and expert SC players. Please rephrase.

Response: Thank you for your suggestion. This sentence has been rephrased.

Change: [P18L321-322] The purpose of this study was to examine the gaze control strategy of the esports experts (Expert) during a game which requires multitasking abilities, by comparing Expert with lower skilled players (Low Skill).

P17L300-301: The authors state ‘The experts showed significantly higher performance scores than the novices in all three Tasks, as expected. Therefore, it seems that the three Tasks modeled successfully discriminated between the abilities of the two subject groups’. This was not the case. Skill level differences could not be found in the easy task. Why might experts and novices differ only for moderate and hard tasks. Discussion should be provided around this.

Response: We apologize for the incorrect statement. We have removed “Easy Task” in this sentence and added a possible reason why there was no significant difference in performance in the Easy Task between expert and low skill players.

Change: [P19L324-325] Since the number of jobs that participants had to perform simultaneously in the Easy Task was small, the Easy Task was likely so easy that it was unable to reveal group differences, probably due to a ceiling effect.

P18L312-313: I would like the authors to speak to their interpretation that ‘the ratio of saccade was larger in the experts than in the novices, suggesting that e-sports experts frequently use saccade for quickly obtaining the information they need.’ This seems to contradict previous literature that suggests information is not gathered during saccades but during fixations. They seem to note this when discussing their fixation ratio findings. I would like the authors to contrast this with expertise literature on quiet eye etc., which would suggest that longer final fixations durations are a marker of expertise.

Response: We appreciate your helpful comment and apologize for the unclear expression that induces a misunderstanding. As you noted, information cannot be gathered during saccades. Thus, we believe that esports experts frequently and effectively use saccade to quickly move the gaze, and judge information at the location where the saccade arrives. We have added the descriptions to contrast our findings in esports with some papers on gaze movements during general sports.

Change: [P19L338-339] Second, the ratio of saccade was larger in the Expert than in the Low Skill players, suggesting that esports experts frequently use saccade to quickly process multiple stimuli. 

Change: [P21L361-373] It is well known that skilled athletes in general sports also show different gaze behavior from novice or low skill players. For example, in one-on-one defensive situations in soccer, novice players mostly watch the ball, while experienced players watch not only the ball but also the knee and hip of the opponent player [5], probably because experienced players pay attention more broadly to the overall movement of their opponent. Similarly, in RTS players must pay attention to multiple stimuli. The current study may expand the knowledge of gaze control strategy during sports, i.e., the strategy of distributing the gaze to wide areas in the visual field might be common when players in either general sports or esports need to pay attention to multiple information streams simultaneously. Meanwhile, it has been proposed that athletes in general sports use different gaze control strategies depending on the situation [26]. For example, elite basketball players fix their gaze on the hoop significantly longer than novices do in a free throw situation [21]. In this case, throwing the ball into the hoop is the only requirement, so it is advantageous for players to fix their gaze on the hoop. From the above, we can estimate that esports players would probably also utilize different gaze strategies depending on the game genre. However, this remains to be elucidated in future studies. 

P18L326: Starcraft is a RTS, not an AVG. Please revise here and throughout.

Response: Thank you for this comment. Now we have changed AVG to RTS in the Discussion section.

P19L331-333: This argument that experts direct focus to the foot in soccer does not support the authors claim that displaying more saccades across a larger horizontal field of view is a marker of expertise. In fact, it suggests the opposite, that experts in traditional sports maintain focus on a small area whereas experts in esports do not. I hope they might be able to clarify and add to their argument here.

Response: We really appreciate your suggestion. We have modified and added explanations regarding the difference and commonality between esports and general sports as follows: 

Change: [P21L361-371] It is well known that skilled athletes in general sports also show different gaze behavior from novice or low skill players. For example, in one-on-one defensive situations in soccer, novice players mostly watch the ball, while experienced players watch not only the ball but also the knee and hip of the opponent player [5], probably because experienced players pay attention more broadly to the overall movement of their opponent. Similarly, in RTS players must pay attention to multiple stimuli. The current study may expand the knowledge of gaze control strategy during sports, i.e., the strategy of distributing the gaze to wide areas in the visual field might be common when players in either general sports or esports need to pay attention to multiple information streams simultaneously. Meanwhile, it has been proposed that athletes in general sports use different gaze control strategies depending on the situation [26]. For example, elite basketball players fix their gaze on the hoop significantly longer than novices do in a free throw situation [21]. In this case, throwing the ball into the hoop is the only requirement, so it is advantageous for players to fix their gaze on the hoop.

FIGURES:

Figure 4: incorrect reporting of statistical findings. Firstly, graphs should be shown as two plots (one comparing Novice to expert with data pooled for difficulty in the two bars – then a significance star between the two bars to show the main effect of skill level. Then the second plot, 3 bars with expertise pooled in each of the easy, moderate and hard bars, with star over easy to show it was different from the other two bars. The vertical graph can be separated into two plots in the same way, such that you have figures 4a, 4b, 4c, and 4d with a and b plots showing horizontal data and c and d plots showing vertical data.

Response: Thank you for your advice on figures. Accordingly, Figure 4 has been separated into two different plots (Skill level and Task difficulty) and the labeling of each plot has been changed to a, b, c, and d. 

Figures 5, 6 and 7 can all be adapted in the same way. If the authors feel strongly about presenting the individual skill level and expertise bars, I would recommend the presentation of the data in this way be done in the form of supplementary material figures. The figures as they are, are not effective in their display of the main findings and support for or against the purpose and hypotheses laid out in this manuscript.

Response: Thank you for your consideration. We think that following your recommendations is a more effective way to explain our findings. Figure 5, 6, and 7 have been edited like Figure 4 was (separated into two plots and a changed labeling method).

Figure 5

Figure 6

Figure 7

Reviewer#2:

General comments: Thank you for your manuscript that investigates the difference in aspects of gaze control between experts esports players and novices while playing the game StarCraft. However, I will address major concerns that could encourage authors to improve the manuscript.

Response: We sincerely appreciate your careful review and helpful comments. We hope the manuscript will be improved with this revision.

First of all, the introduction section needs some rework, starting from implementing or acknowledging the current standardise definitions and conceptualisation of esports. This, as authors are using a wrong spelling of the word esports and it is important to clarify that esports is not a subcategory of action video games (AVG) (see review, (Pedraza-Ramirez et al., 2020).

Response: Thank you for your suggestion. Definitions and conceptualization of esports have been edited. Additionally, we are sorry for spelling esports incorrectly. Now, “e-sports” has been changed to “esports” and “Esports”. We understand that esports is not a subcategory of action video games. We have shown the difference between them as follows:

Change: [P2L35-36] Irrespective of game genre, when a game is played competitively and professionally, it can be considered esports [2].

Even though authors are acknowledging the current literature of video games and cognitive functions, it is recommended to consider including not only the single studies in the video gaming literature but as well the recent meta-analysis that have found mixed evidence on the effects of video game practice on cognition (e.g., Bediou et al., 2018; Sala et al., 2017) and the current state of the art and relevant studies in esports (e.g., Li et al., 2020; Pedraza-Ramirez et al., 2020; Thompson et al., 2019; Toth et al., 2019).

Response: Thank you for your suggestion about references. We carefully checked your reference list and decided to add the references about meta-analysis of esports (Li et al. 2020) and related research about esports (Bediou et al. 2018). 

Change: [P3L40-44] For example, Kowal et al. [16] found that the task-switching ability of action video game (AVG) players is higher than that of non-AVG players. Furthermore, top video game players have better cognitive flexibility than novice players [15] in their ability to adapt to new and unexpected conditions [7]. In addition, a recent meta-analysis also found that video game practice has positive effects on cognitive function [14] (the mental processing required to obtain information and knowledge [6]).

In general, the manuscript needs to improve at the theoretical level. This, because it was not mentioned any theoretical background that can help to support the aim of the study and develop clear assumptions. Authors identified different areas of investigation in relation to the aim of the study such as cognitive functions, multitasking or motor-cognitive processes, however, it is not clear the theoretical foundation. However, due to the lack of specificity the manuscript is not doing a good job showing how this research will contribute to the extension of knowledge. Even though it was shown in the manuscript a great number of studies supporting the importance to study visual functions and saccade control ability in video game players, authors need to consider, first, a theoretical account that will help to support the hypothesis. Second, acknowledging the considerations towards the issues of lack of rigorous methodological designs in video game research and esports (e.g., not grouping video game genres) (see Dale & Green, 2017). Third, differentiating and bring specificity whether the study is on esports or video games. Consequently, the current state of the hypothesis is too vague and do not have a supporting evidence, thus, affecting the trustworthiness of the study. 

We are thankful for your comment. We have tried to refocus the theoretical underpinnings of our manuscript and have added some comments on how this study contributes to an extension of our knowledge.

Response to the first point (a theoretical account that will help to support the hypothesis): 

Thank you for your suggestion. We have modified the sentences explaining the theoretical background for setting the hypothesis as follows:

Change: [P3L45-P4L61] The real-time strategy (RTS) game is another kind of video game. In RTS games players are required to multi-tasking using complex strategies. AVG and RTS have similar positive effects on cognitive processes (both AVG and RTS improve reaction time and problem-solving ability) [10]. Meanwhile, Dobrowolski et al. [4] found that RTS players have superior cognitive function compared to that of AVG players. Indeed, RTS game training improves task switching ability [18]. In order to switch between tasks, it is necessary to accurately perceive stimuli presented on the monitor screen through proper gaze control. 

Indeed, in RTS, multiple information streams appear simultaneously in different locations on the monitor screen, and players must absorb and select information of importance to them. Based on the information obtained, they must judge how to make an appropriately timed response to the stimuli. Thus, multitasking ability could be the key element underlying higher performance in RTS. RTS players must move their gaze over multiple areas on the monitor to win the game. Therefore, esports players would use more ballistic gaze movements, called saccade, to enable them to process multiple stimuli more quickly. However, the characteristics of gaze strategy in esports players are still unknown. Therefore, the purpose of the current study was to examine the gaze control of RTS expert players. We hypothesized that RTS experts in esports would show superior gaze control strategy (wide and fast saccadic gaze movement) which is evaluated by measuring gaze distribution and saccadic movements. Clarifying the gaze control strategy used by successful esports players would be beneficial for esports coaching, and also for expanding esports markets. 

Response to the second point (lack of rigorous methodological design):

Thank you for advice about methodological designs in video game research and esports. Unfortunately, we did not collect the StarCraft players’ play history of another genre of video game. Thus, we cannot rule out the possibility that other esports play history have created the difference between the Expert and the Low Skill players. This information has been added in the limitation section as follows:

Change: [P22L394-396] In this study, we did not record the participants’ experience in playing other esports or their years of esports experience. Therefore, we cannot rule out the possibility that a history of playing other esports affected the difference in gaze movement between the Expert and the Low Skill players.

Response to the third point (differentiating and bringing specificity to whether the study is on esports or video games.):

We apologize for unclear expressions. The focus of the current study is how the ability and strategy in high level esports players differ from those of people who just enjoy video games. 

Response to all three comments:

In response to your opinion that our hypothesis is too vague, we have modified the description of how the hypothesis was created as follows: 

Change: [P3L51-P4L61] Indeed, in RTS, multiple information streams appear simultaneously in different locations on the monitor screen, and players must absorb and select information of importance to them. Based on the information obtained, they must judge how to make an appropriately timed response to the stimuli. Thus, multitasking ability could be the key element underlying higher performance in RTS. RTS players must move their gaze over multiple areas on the monitor to win the game. Therefore, esports players would use more ballistic gaze movements, called saccade, to enable them to process multiple stimuli more quickly. However, the characteristics of gaze strategy in esports players are still unknown. Therefore, the purpose of the current study was to examine the gaze control of RTS expert players. We hypothesized that RTS experts in esports would show superior gaze control strategy (wide and fast saccadic gaze movement) which is evaluated by measuring gaze distribution and saccadic movements. Clarifying the gaze control strategy used by successful esports players would be beneficial for esports coaching, and also for expanding esports markets. 

Additionally, the method section needs rework, as the procedure is not clearly explained step by step for replicability and is missing important information. For example, we do not know whether was controlled that participants were engaging in other esports or video games or if it was asked for the years of experience playing StartCraft or other games. This, as experience and the type of game played may be crucial elements when investigating expertise differences in relation to cognitive abilities (see Li et al., 2020). Also, authors are not showing an a priori G* Power calculation to estimate the sample size required. Thus, clarifying and addressing the reliability of the methods used, clarity in the participants' background, and the procedure of the data collection brings rigour and reliability of the methodological design. Although the experimental task is very innovative and can add methodological value to research and its applicability in esports due to its ecological validity, I am wondering whether the test is supported by a previous investigation or if it was previously piloted, and if so, it would be required to address the findings so that there are not risks of construct validity. Lastly, I think that they behavioural data could be interesting form a motor learning and motor control perspective, however, I am wondering how this data aligns with the aim of the study. Therefore, integrating the results from this performance variable into the characteristics of gaze control could bring a better understanding of expertise in this manuscript.

Response to the first point (replicability and missing information):

Thank you for your comments. We agree that detailed information is lacking in the methods section and have added information about the participants as follows:

Change: [P22L394-396] In this study, we did not record the participants’ experience in playing other esports or their years of esports experience. Therefore, we cannot rule out the possibility that a history of playing other esports affected the difference in gaze movement between the Expert and the Low Skill players.

Response to the second point (result about G*power analysis):

We apologize for omitting information about the G*power analysis. We calculated the G*power using preliminary experimental data which compared gaze movement of esports expert and low skill players. Specific information about the G*power calculation has been added as follows:

Change: [P5L78-84] Before starting the experiment, we performed a power analysis to estimate the required sample size (RStudio version 1.3). The sample size was calculated using preliminary experimental data related to the distribution of gaze movements of esports experts (Expert) and players with lower skills (Low Skill) (effect size: 0.51; α level: 0.05; power (1-β error probability): 0.80). As a result, the calculated necessary number of participants was seven in each group. According to the result of this power analysis, sixteen participants (15 male, 1 female; 7 Expert, 9 Low Skill; mean age, 22.4 years; age range, 18-28 years) with experience playing StarCraft participated in the present study.

Response to the third point (construct validity 1):

Thank you for your concern about the construct validity of our study. The tasks used in the current study are original and are supported by our preliminary investigations and previous research about RTS (Dobrowolski et al., 2014 and Basak et al., 2014). All participants agreed that our tasks required a high level of multitasking ability. Information about preliminary investigations and our reason for using an original task have been added as follows:

Change: [P4L62-67] Murphy and Spencer [17] argue that there was no significant difference between esports players and non-esports players in visual attention skill; this conclusion was verified by using only a simple attention blink task in a laboratory, which was unrealistic because there was no multitasking involved. Therefore, in the current study for analyzing how esports players control their gaze while playing a game, we set them a task which was similar to what occurs in real RTS games; we required players to exhibit higher multitasking abilities, and to cope with increasing task difficulty. 

Change: [P10L168-170] Each Task was original and conducted in response to preliminary investigations measuring gaze movement in esports experts. Participants in our preliminary investigations agreed that a high level of multitasking abilities was required for each Task.

Response to the fourth consideration (construct validity 2):

The reason why we used our original task is that previous research papers have only verified the cognitive function in RTS players by using simple laboratory tasks. To be specific, Murphy and Spencer (2009) argue that there was no significant difference between esports players and non esports players in visual attention skill. However, visual attention skill was only verified using a simple attention blink task. Thus, we thought it necessary to focus on the gaze movement of RTS players in actual video game circumstances with a higher difficulty level and closer resemblance to reality. 

Change: [P3L54-P4L67] RTS players must move their gaze over multiple areas on the monitor to win the game. Therefore, esports players would use more ballistic gaze movements, called saccade, to enable them to process multiple stimuli more quickly. However, the characteristics of gaze strategy in esports players are still unknown. Therefore, the purpose of the current study was to examine the gaze control of RTS expert players. We hypothesized that RTS experts in esports would show superior gaze control strategy (wide and fast saccadic gaze movement) which is evaluated by measuring gaze distribution and saccadic movements. Clarifying the gaze control strategy used by successful esports players would be beneficial for esports coaching, and also for expanding esports markets. 

Murphy and Spencer [17] argue that there was no significant difference between esports players and non-esports players in visual attention skill; this conclusion was verified by using only a simple attention blink task in a laboratory, which was unrealistic because there was no multitasking involved. Therefore, in the current study for analyzing how esports players control their gaze while playing a game, we set them a task which was similar to what occurs in real RTS games; we required players to exhibit higher multitasking abilities, and to cope with increasing task difficulty. 

Response to the fifth consideration (relationship between behavioral data and gaze movement): 

Thank you for the comment. As described in the manuscript, our results suggest that esports experts show superior ability both from behavioral data (game performance) and gaze control parameters (horizontal gaze distribution, saccade percentage, saccade velocity, and saccade number). As for the relationship between them, some gaze control parameters showed significant correlation with behavioral data, but this significant correlation was not observed in every parameter, probably due to the small sample size. Please check the following graphs. 

Finally, it is necessary to strengthen and highlight the added values and extension of knowledge at the introduction and support it in the discussion. The discussion section will benefit of integrating supporting evidence align with the findings. I think authors are doing a good job in lines 324 to 339 by discussing the results with previous evidence and expanding the knowledge.

The comments are mean to encourage authors to improve the quality and specificity of the direction of the manuscript.

Response: Thank you for your positive comment about our manuscript. We believe our revised manuscript is stronger than the original, and we highlight our findings as follows:

Change: [P21L361-373] It is well known that skilled athletes in general sports also show different gaze behavior from novice or low skill players. For example, in one-on-one defensive situations in soccer, novice players mostly watch the ball, while experienced players watch not only the ball but also the knee and hip of the opponent player [5], probably because experienced players pay attention more broadly to the overall movement of their opponent. Similarly, in RTS players must pay attention to multiple stimuli. The current study may expand the knowledge of gaze control strategy during sports, i.e., the strategy of distributing the gaze to wide areas in the visual field might be common when players in either general sports or esports need to pay attention to multiple information streams simultaneously. Meanwhile, it has been proposed that athletes in general sports use different gaze control strategies depending on the situation [26]. For example, elite basketball players fix their gaze on the hoop significantly longer than novices do in a free throw situation [21]. In this case, throwing the ball into the hoop is the only requirement, so it is advantageous for players to fix their gaze on the hoop. From the above, we can estimate that esports players would probably also utilize different gaze strategies depending on the game genre. However, this remains to be elucidated in future studies.

---

## [Decision Letter · Decision Letter 1]

11 Oct 2021

PONE-D-21-09912R1Gaze control ability in esports expertsPLOS ONE

Dear Dr. Nakagawa,

Thank you for submitting your manuscript to PLOS ONE. After careful consideration, we feel that it has merit but does not fully meet PLOS ONE’s publication criteria as it currently stands. Therefore, we invite you to submit a revised version of the manuscript that addresses the points raised during the review process.

We look forward to receiving your revised manuscript.

Kind regards,

Greg Wood, PhD

Academic Editor

PLOS ONE

Reviewers' comments:

Reviewer's Responses to Questions

**Comments to the Author**

1. If the authors have adequately addressed your comments raised in a previous round of review and you feel that this manuscript is now acceptable for publication, you may indicate that here to bypass the “Comments to the Author” section, enter your conflict of interest statement in the “Confidential to Editor” section, and submit your "Accept" recommendation.

Reviewer #1: (No Response)

Reviewer #2: (No Response)

2. Is the manuscript technically sound, and do the data support the conclusions?

Reviewer #1: Yes

Reviewer #2: Partly

3. Has the statistical analysis been performed appropriately and rigorously? 

Reviewer #1: Yes

Reviewer #2: Yes

4. Have the authors made all data underlying the findings in their manuscript fully available?

Reviewer #1: No

Reviewer #2: No

5. Is the manuscript presented in an intelligible fashion and written in standard English?

Reviewer #1: Yes

Reviewer #2: No

6. Review Comments to the Author

Reviewer #1: Many thanks to the authors for addressing my initial comments.

The paper has improved significantly, but I do feel requires a number of revisions and amendments still before meriting publication.

My comments are listed below.

Title:

As your paper address the difference in gaze control ability between low and high skill RTS players, the title should be revised to reflect this more accurately. I suggest ‘Gaze control ability between low and high skill players of a real-time strategy game.

Abstract:

You often interplay between multitasking and task-switching ability throughout the manuscript. I would argue that your data are attempting to support the latter as humans generally cannot multitask and what we might think as multitasking has been shown to just be evidence of superior task switching ability. I would suggest the authors replace multitasking with task-switching throughout the manuscript.

Introduction

Line 36: Esports are the competitive and often professional play. They don’t have to be professional to be esports. Revise the sentence to amend the wording please.

Line 45-46: Change multitasking to multitask.

Lines 46-47: Change ‘(both AVG and RTS improve reaction time and problem-solving ability)’ to ‘(playing either AVG and RTS games improves reaction time and problem-solving ability)’

Line 55: Change saccade to saccades.

Line 58: By ‘wide’ do you mean dispersed? This might be a better word. Also, I would assume it is the fixations and not saccades over a larger area that is a marker of expertise? So faster saccades and a larger area over which fixations are dispersed? If so, please amend.

Lines 60-61: why would clarifying the gaze control strategy used by successful esports players be beneficial for expanding esports markets? Please clarify in the manuscript.

Lines 62-67: This paragraph creates confusion as everything you have leading up to the study’s purpose centres around multitasking/task switching ability. I understand that attention is relevant here especially when looking t fixations, but this paragraph needs to come earlier and be part of the setup to the purpose, not after it. Please amend.

Line 68: Starcraft needs to be introduced earlier. Again, everything should be setup before the purpose so when it is introduced at the very end of the introduction section, it seems obvious to the reader that the purpose is what you say it is, then follow the purpose with your hypotheses to conclude your introduction.

Methods

Lines 78-84: Please provide your power analysis in your reply here. Why use Rstudio and not Gpower, which is the typical standard. Given you are using prelim experimental data, I assume you have means and SD’s you are inputting for your calculations? Please provide these. Finally, your power analysis should be outputting equal sample sizes between your groups which may or may not match what you have collected exactly. As such, I don’t believe the power analysis was conducted properly and I think a supplementary file or some file here demonstrating the raw inputs and outputs from the power analysis is needed to verify the study is appropriately powered to detect the differences found.

Line 97: Same comment as above. The jobs are not performed simultaneously but require very rapid and accurate task switching ability. Please amend.

Line 127-128: Watch grammar and make sure tense is kept consistent. The sentence “The difficulty of each Task depends on how many jobs are required and how many Zones are in play at the same time” should read “The difficulty of each Task depended on how many jobs were required and how many Zones were in play at the same time”. This is important and needs to be proof-read and corrected throughout the manuscript to improve readability.

Line 177-178: The equation you present equally weights the sum of buildings constructed and units produced, and the enemy units destroyed. To more strongly emphasize the units destroyed, sum the two of these together but multiply each by a constant that is larger for enemy units destroyed.

i.e., (units produced + buildings constructed)*A + (enemies destroyed)*B, where B > A

Lines 236-237: Change ‘calculated’ to analysed.

Line 237-239: change “When the Kruskal-Wallis test detected significance, the Wilcoxon Rank-Sum test was performed to find the difference between each group of Task difficulties (Bonferroni’s correction (p < .05 divided by 3 tests: significance threshold at .017)).”

To

“When the Kruskal-Wallis test detected significance, the Wilcoxon Rank-Sum test was performed to determine which Task difficulties were significantly different from one another (Bonferroni’s correction (p < .05 divided by 3 tests: significance threshold at .017)).”

Results

Lines 299-302: you report finding main effects for skill and task difficulty, with no interaction. However, your post hoc reports an interaction in that low skill participants had higher fixation percentage in the moderate compared to easy task. Please revise your reporting here as the main effect suggests you should only be reporting expertise differences whereby all task difficulties are pooled, OR, reporting differences between task difficulties, whereby all expertises within each task difficulty are pooled.

Discussion

Line 333-334: Please change

‘Thus, when playing such games, it is necessary to simultaneously pay attention to multiple points and areas on the screen.’

to

“Thus, when playing such games, it is necessary to rapidly switch attention to multiple points and areas on the screen and maintain this information in working memory.’

Line 338: Saccades are never used to process information. Information is gathered during fixations. Please amend the wording of this sentence to reflect this.

Line 346: The way this is written is counterintuitive. If Experts exhibit more saccades, they would also have more fixations. In your case, this is also true, but low skill individuals have a higher fixation time. So to say experts have more saccades but low skill players have a higher percentage of fixation could imply that low skill players have more fixations. Please amend wording to clarify that it is in fact that low skill individuals have longer fixation durations.

Line 352: Change multitasking to task switching.

Reviewer #2: General Comments

Thank you for resubmitting your manuscript and consideration of my general comments. The authors have addressed many of the suggestions, however, I will address some concerns that I still think need further attention.

Overall, the manuscript presents improvements in the different sections where concerns were raised. However, the introduction still needs to be revisited as it reads too general and does not present a clear theoretical framework. A theory driven study will help the authors to propose hypotheses based on evidence and give clear direction to the further of knowledge.

Additionally, there are a few inconsistencies while describing real time strategy games and action video games, esports and video games especially when referring to previous work (see specific comments). Furthermore, while the aim of the study is to examine gaze control of RTS expert players, first, it would be important to emphasise that the aim is directed towards StarCraft players and not the whole RTS genre; Second, the role of cognitive abilities needs to be clarified, more precisely those with greater importance for the aim of the study require further detail and argumentation. Even though the authors are briefly addressing different studies showing the benefits of esports on specific cognitive abilities, it is hard to follow the red threat on why multitasking is an important ability for esports performance and what is the connection with gaze behaviour. Consequently, the theoretical link and underlying mechanisms at the specific cognitive level are still missing.

Furthermore, there are still issues with justifying the sample size and the validity and reliability of the task. I have noted a few specific comments on these issues below. Lastly, it is important to review the writing in some sections and the use of capital letters where it shouldn’t be.

Specific Comments

P3L45: Real time-strategy is not a video game genre and not a video game itself.

P3 L45-47: Please rework this paragraph

P3 L51-56: If the main arguments and justifications of the study are going in the direction of the role and importance of multitasking in RTS games, it is necessary to first, extend on the literature of multitasking and second, support these arguments with evidence. So far this section still not strongly supported by empirical evidence and lacks theoretical accounts that can help to justify this link with video games and esports.

P 3 L58-61: The hypothesis does not seem to have a theoretical link still justifying why authors are expecting those findings. However, this hypothesis reads very simplistic, as it is well known that experts will outperform novices in specific tasks. Thus, the hypothesis in its current state does not add knowledge and still too vague.

P3 L60-61: The practical implications as mentioned here for esports coaching and expanding esports markets should be either supported in the introduction with theoretical justifications or addressed at the end of the manuscript supported by the findings.

P3 L66-67: As the test seems to fulfil some elements of ecological validity, I would expect justification why multitasking and the increase of task difficulty can help to understand closer what the StarCraft players experience while playing the game. Consequently, support from ecological validity and representative designs studies could help to support the decisions on the difficulty of the task.

P3 L62-64: The investigation used as support for the current study Murphy and Spencer (2009) was a replication study from Green and Bavelier (2003) in which the sample were not esports players as you mention and the games, they are focusing on are not RTS but action games in which combines different games. Thus, this argumentation loses strength and validity when the full story is not explained. The aim of these previous studies should mention and fully considered when comparing the results, aiming to extends your research based on those findings.

P3 L64-67: I am not sure if this is adding to the hypothesis. It seems that the authors are interested in investigating multitasking and coping mechanisms. The multitasking bit was mentioned earlier but the introduction lacks literature in this field and its importance for the study.

P5 L75: Please be consistent with the language or make sure that connection between gaze control and multitasking is clear. So far, it is confusing to understand which are/is the real aim of the study, as there are different concepts drop during the introduction and methodology. Thus, the theoretical account is not clear, and hypothesis are not supported theoretically or empirically.

P5 L78-81: Which preliminary experimental data are you referring to? This needs to be further justified.

P5 L84-91: The are still information missing. For example, the recruitment strategy and the inclusion/exclusion criteria are still missing.

P9 L152-153: I think that the authors should allocate a section where the procedure of the experiment is explained in detailed. Additionally, a figure could help to show the overview of the experimental procedure and measures. This, as I see a critical issue in terms of the procedure of the study as there was not randomization of the difficulty of the tasks.

P10 L167-169: The preliminary investigations that you are referring to support the sample size and the reliability and validity of the task need to be further described and adequately referenced.

P11 L189: How did you control for the use of different settings of keyboard and mouse? And how this decision could influence the results of the test.

7. PLOS authors have the option to publish the peer review history of their article (what does this mean?). If published, this will include your full peer review and any attached files.

Reviewer #1: No

Reviewer #2: No

---

## [Author Response · Author response to Decision Letter 1]

23 Nov 2021

Response to Reviewers

Dear Reviewers

Thank you very much for your feedback, comments, and suggestions. Our responses to your comments are as follows:

Reviewer #1:

General comment: Many thanks to the authors for addressing my initial comments.

The paper has improved significantly, but I do feel requires a number of revisions and amendments still before meriting publication. My comments are listed below.

Response: Thank you for your general comment. We hope we have improved our manuscript through this revision.

Title: As your paper address the difference in gaze control ability between low and high skill RTS players, the title should be revised to reflect this more accurately. I suggest ‘Gaze control ability between low and high skill players of a real-time strategy game.

Response: Thank you for your suggestion. We decided to follow your suggestion about the title. Thus, we changed the title ‘Gaze control ability in esports expert’ to ‘Difference in gaze control ability between low and high skill players of a real-time strategy game in esports’.

Change: [P1L3-4] Difference in gaze control ability between low and high skill players of a real-time strategy game in esports.

Abstract: You often interplay between multitasking and task-switching ability throughout the manuscript. I would argue that your data are attempting to support the latter as humans generally cannot multitask and what we might think as multitasking has been shown to just be evidence of superior task switching ability. I would suggest the authors replace multitasking with task-switching throughout the manuscript.

Response: We appreciate your comments. We changed the word ‘multitasking ability’ to ‘task-switching ability’ in the abstract and throughout the manuscript.

Change: [P2L28-29] This could be one of the factors enabling esports experts to perform better than players with lower skills when playing games that require task-switching ability.

Introduction:

Line 36: Esports are the competitive and often professional play. They don’t have to be professional to be esports. Revise the sentence to amend the wording please.

Response: Thank you for your comment. We revised the wording in this sentence.

Change: [P2L35-36] Irrespective of their genre, esports are very competitive and often professionally played.

Line 45-46: Change multitasking to multitask.

Response: Thank you. The word ‘multitasking’ has been changed to ‘task-switching’ and is used consistently throughout the manuscript.

Change: [P3L46] In an RTS game, players are required to task-switch using complex strategies,

Lines 46-47: Change ‘(both AVG and RTS improve reaction time and problem-solving ability)’ to ‘(playing either AVG and RTS games improves reaction time and problem-solving ability)’

Response: Thank you for your suggestion. The sentence in this section has been entirely changed.

Change: [P3L49-51] It is probable that considering these characteristics of an RTS game, positive effects of playing RTS games on cognitive processes (reaction time and problem-solving ability) occur [10].

Line 55: Change saccade to saccades

Response: We appreciate the suggestion. The word ‘saccade’ has been changed to ‘saccades’.

Change: [P4L62] saccades, 

Line 58: By ‘wide’ do you mean dispersed? This might be a better word. Also, I would assume it is the fixations and not saccades over a larger area that is a marker of expertise? So faster saccades and a larger area over which fixations are dispersed? If so, please amend.

Response: Thank you for your consideration. We assumed that a larger fixation area is a marker of expertise. Thus, the wording about ‘wide’ has been changed to ‘dispersed’.

Change: [P4L63-64] (dispersed and fast saccadic gaze movement).

Lines 60-61: why would clarifying the gaze control strategy used by successful esports players be beneficial for expanding esports markets? Please clarify in the manuscript.

Response: Thank you for your opinion. We deleted the sentence which mentioned expanding the esports market, because it has essentially no relationship to our research. Also, the sentence about esports coaching has been moved to the discussion section.

Change: [P22L392-393] Additionally, knowledge about the gaze control strategy used by successful esports players would be helpful for esports coaching and for devising new training methods.

Lines 62-67: This paragraph creates confusion as everything you have leading up to the study’s purpose centers around multitasking/task switching ability. I understand that attention is relevant here especially when looking t fixations, but this paragraph needs to come earlier and be part of the setup to the purpose, not after it. Please amend.

Response: Thank you for your suggestion. The content of lines 62-67 in the previous version has been removed because another reviewer claimed that this previous research lessened the strength and validity of the current study. Additionally, a new hypothesis and background of our research has been added (P3L46-P4L65).

Line 68: Starcraft needs to be introduced earlier. Again, everything should be setup before the purpose so when it is introduced at the very end of the introduction section, it seems obvious to the reader that the purpose is what you say it is, then follow the purpose with your hypotheses to conclude your introduction.

Response: We entirely agree with your opinion. We introduce StarCraft earlier and it is now explained before the purpose section (P3L46-P4L65).

Method:

Lines 78-84: Please provide your power analysis in your reply here. Why use Rstudio and not Gpower, which is the typical standard. Given you are using prelim experimental data, I assume you have means and SD’s you are inputting for your calculations? Please provide these. Finally, your power analysis should be outputting equal sample sizes between your groups which may or may not match what you have collected exactly. As such, I don’t believe the power analysis was conducted properly and I think a supplementary file or some file here demonstrating the raw inputs and outputs from the power analysis is needed to verify the study is appropriately powered to detect the differences found.

Response: Thank you for your comment. There was no specific reason why we used RStudio. Thus, we calculated the required sample size using G*Power software. G*Power was calculated by using the Hard Task performance level of experts (Expert, n = 4) and players with lower skills (Low Skill, n = 4) of our preliminary experiment. In the main experiment we used the different sample size for the two groups, because it was easy to recruit Low Skill subjects. Raw input and output of the power analysis is as follows:

Raw data of each participant: Expert (224, 212, 153, 225), Low Skill (146, 176, 113,141)

Power calculation for unpaired t-test between Expert (n = 4) vs. Low Skill (n = 4)

t test: Means: Difference between two independent means (two groups)

Analysis: Compute required sample size

Input: Tail(s) = Two

 Effect size d = 1.91

 α err prob = 0.05

 Power (1-β err prob)) = 0.9

 Allocation ration N2/N1 = 1

Output: Noncentrality parameter δ = 3.57

 Critical t = 2.17

 Df = 12

 Sample size group 1 = 7

 Sample size group 2 = 7

 Total sample size = 14

 Actual power = 0.905

Specific G*Power protocols have been provided in Supplement file S6. 

Change: [P4L72-P5L77] We performed a power analysis to estimate the required sample size (G*Power version 3.1). The G*Power was calculated using the Hard Task performance level (described in the Sample games section) of experts (Expert, n = 4) and the performance of players with lower skills (Low Skill, n = 4) in a preliminary experiment (Cohen’s d: 1.91; α level: 0.05; power (1-β error probability): 0.9). Effect size was calculated according to Cohen [17]. The results showed that seven participants were required for each group. The details of the G*Power protocol are presented in the S6 Supplement file.

Line 97: Same comment as above. The jobs are not performed simultaneously but require very rapid and accurate task switching ability. Please amend.

Response: Thank you for this comment. The word ‘multitasking’ has been changed to ‘task-switching’.

Change: [P6L103-104] In this process a high level of task-switching ability is required to perform all these jobs simultaneously.

Line 127-128: Watch grammar and make sure tense is kept consistent. The sentence “The difficulty of each Task depends on how many jobs are required and how many Zones are in play at the same time” should read “The difficulty of each Task depended on how many jobs were required and how many Zones were in play at the same time”. This is important and needs to be proof-read and corrected throughout the manuscript to improve readability.

Response: Sorry for grammatical error of our manuscript. We revised the content of Line127-128 and the whole manuscript. 

Change: [P8L133-135] The difficulty of each task depended on how many jobs were required and how many Zones were in play at the same time. When more jobs were required and more Zones were used at the same time, the difficulty of the task increased.

Line 177-178: The equation you present equally weights the sum of buildings constructed and units produced, and the enemy units destroyed. To more strongly emphasize the units destroyed, sum the two of these together but multiply each by a constant that is larger for enemy units destroyed.

i.e., (units produced + buildings constructed)*A + (enemies destroyed)*B, where B > A

Response: Thank you for suggestion. We followed your suggestion and reworked the performance level.

Change: [P11L178-179] (number of constructed buildings + number of produced units) ×2+ (number of destroyed enemy units) ×3

The reason for using different constants is to strongly emphasize the number of destroyed enemy units.

Change: [P15L255-257] (Moderate Task: mean 170.28 ± SD 19.20 vs. mean 137.22 ± SD 25.97, t = 3.31, df = 15.89, p = .004, and Hard Task: mean 208.14 ± SD 26.01 vs. mean 155.55 ± SD 37.26, t = 3.75, df = 15.71, p = .001)

Lines 236-237: Change ‘calculated’ to analyzed.

Response: Thank you. The word ‘calculated’ has been changed to ‘analyzed’

Change: [P14L237-238] For this reason, these data were analyzed by a non-parametric method.

Line 237-239: change “When the Kruskal-Wallis test detected significance, the Wilcoxon Rank-Sum test was performed to find the difference between each group of Task difficulties (Bonferroni’s correction (p < .05 divided by 3 tests: significance threshold at .017)).”

To

“When the Kruskal-Wallis test detected significance, the Wilcoxon Rank-Sum test was performed to determine which Task difficulties were significantly different from one another (Bonferroni’s correction (p < .05 divided by 3 tests: significance threshold at .017)).”

Response: Thank you for this correction. The sentence ‘find the difference between each group of Task difficulties’ has been changed to ‘determine which Task difficulties were significantly different from one another’.

Change: [P14L238-L240] When the Kruskal-Wallis test detected significance, the Wilcoxon Rank-Sum test was performed to determine which Task difficulties were significantly different from one another (Bonferroni’s correction (p < .05 divided by 3 tests: significance threshold at .017)).

Result:

Lines 299-302: you report finding main effects for skill and task difficulty, with no interaction. However, your post hoc reports an interaction in that low skill participants had higher fixation percentage in the moderate compared to easy task. Please revise your reporting here as the main effect suggests you should only be reporting expertise differences whereby all task difficulties are pooled, OR, reporting differences between task difficulties, whereby all expertises within each task difficulty are pooled.

Response: We apologize for our statistical error. Our statistics reporting has been edited throughout the entire manuscript. 

Change: [P15L268-P16L270] Simple main effect analysis found that SD of horizontal gaze distribution during the Easy Task was higher than that during the Moderate Task and Hard Task, irrespective of skill level (p < .001 for both).

Change: [P17L301-303] Simple main effect analysis indicated that the fixation percentage in the Moderate Task was significantly higher than in the Easy Task, irrespective of skill level (p = .02).

Line 333-334: Please change

‘Thus, when playing such games, it is necessary to simultaneously pay attention to multiple points and areas on the screen.’

to

“Thus, when playing such games, it is necessary to rapidly switch attention to multiple points and areas on the screen and maintain this information in working memory.’

Response: Thank you for this correction. The text has been corrected as pointed out.

Change: [P19L334-336] Thus, when playing such games, it is necessary to rapidly switch attention between multiple points and areas on the screen and maintain this information in working memory.

Line 338: Saccades are never used to process information. Information is gathered during fixations. Please amend the wording of this sentence to reflect this.

Response: Thank you for this correction. We understand saccades are never used to process information. Thus, we argued that saccades are used to pay attention to multiple points and areas on the screen. 

Change: [P19L338-341] Second, the ratio of saccade was larger and fixation time was shorter in the Expert than in the Low Skill players, suggesting that esports experts frequently use saccade to quickly change their gaze position and accumulate more information through their shorter fixation time. 

Line 346: The way this is written is counterintuitive. If Experts exhibit more saccades, they would also have more fixations. In your case, this is also true, but low skill individuals have a higher fixation time. So to say experts have more saccades but low skill players have a higher percentage of fixation could imply that low skill players have more fixations. Please amend wording to clarify that it is in fact that low skill individuals have longer fixation durations.

Response: Thank you for this advice. We amend the wording to clarify that the Low Skill players have longer fixation durations.

Change: [P20L348-350] Finally, the higher percentage of fixation in the Low Skill than in the Expert players (Figure 7c and 7d) indicates that the Low Skill players needed longer fixation duration to get information and therefore needed to keep their gaze on one location longer.

Line 352: Change multitasking to task switching.

Response: Thank you. Multitasking has been changed to task-switching.

Change: [P20L354-355] These eye movements are advantageous to processing multiple stimuli and successfully playing games when task-switching ability is required.

Reviewer #2:

General comment: Thank you for resubmitting your manuscript and consideration of my general comments. The authors have addressed many of the suggestions, however, I will address some concerns that I still think need further attention.

Response: Thank you for your careful review and your advice. Our manuscript has been reworked according to your specific comments.

Overall, the manuscript presents improvements in the different sections where concerns were raised. However, the introduction still needs to be revisited as it reads too general and does not present a clear theoretical framework. A theory driven study will help the authors to propose hypotheses based on evidence and give clear direction to the further of knowledge.

Response: Thank you for your consideration. The Introduction section has been reworked according to your suggestions. Please check the revised Introduction parts, especially from the second to the fourth paragraph (P3L45-P4L65). 

Additionally, there are a few inconsistencies while describing real time strategy games and action video games, esports and video games especially when referring to previous work (see specific comments). 

Response: Thank you for comments on the usage inconsistences of RTS, video game, and esports. We checked through the manuscript and corrected these terms.

Furthermore, while the aim of the study is to examine gaze control of RTS expert players, first, it would be important to emphasise that the aim is directed towards StarCraft players and not the whole RTS genre; 

Response: Thank you for this suggestion. We emphasized that this study analyzed StarCraft players to shed light on the characteristic of gaze movement in RTS players.

Change: [P3L55-57] In light of the above background, the current study aimed to clarify how RTS players control their gaze while playing a game. In order to shed light on their gaze control strategy, we adopted a popular RTS game, StarCraft, as the model task. 

Second, the role of cognitive abilities needs to be clarified, more precisely those with greater importance for the aim of the study require further detail and argumentation. Even though the authors are briefly addressing different studies showing the benefits of esports on specific cognitive abilities, it is hard to follow the red threat on why multitasking is an important ability for esports performance and what is the connection with gaze behaviour. Consequently, the theoretical link and underlying mechanisms at the specific cognitive level are still missing.

Response: Thank you for this suggestion. We reworked the whole introduction section to explain the reasons for focusing on the gaze control ability and multitasking ability (we have changed “multitasking ability” to “task-switching ability” according to another reviewer’s comments).

Furthermore, there are still issues with justifying the sample size and the validity and reliability of the task. I have noted a few specific comments on these issues below. 

Response: Thank you for your consideration. The sample size has been recalculated with G*Power software. We provide a detailed description regarding the protocol of this power analysis in the revised manuscript. We believe that this information can increase the validity and reliability of our research.

Change: [P4L72-P5L77] We performed a power analysis to estimate the required sample size (G*Power version 3.1). The G*Power was calculated using the Hard Task performance level (described in the Sample games section) of experts (Expert, n = 4) and the performance of players with lower skills (Low Skill, n = 4) in a preliminary experiment (Cohen’s d: 1.91; α level: 0.05; power (1-β error probability): 0.9). Effect size was calculated according to Cohen [17]. The results showed that seven participants were required for each group. The details of the G*Power protocol are presented in the S6 Supplement file.

Lastly, it is important to review the writing in some sections and the use of capital letters where it shouldn’t be.

Response: Thank you for your suggestion. ‘Task difficulty’ has been changed to ‘task difficulty’ in whole manuscript. Additionally, ‘Task’ has been changed to ‘task’ where the word needed to be revised. Additionally, ‘Screen Marker Calibration’ and ‘Eye Movement Detector’ have been changed to ‘screen marker calibration’ and ‘eye movement detector’

P3L45: Real time-strategy is not a video game genre and not a video game itself.

Response: Thank you for this consideration. We removed the description that Real-time strategy (RTS) is a video game genre.

P3 L45-47: Please rework this paragraph

Response: Thank you for your comment. This paragraph has been reworked.

Change: [P3L45-48] The real-time strategy (RTS) game is a popular esports genre. In an RTS game, players are required to task-switch using complex strategies, because multiple information streams appear simultaneously in different locations on the monitor screen, and players must absorb and select information of importance to them.

P3 L51-56: If the main arguments and justifications of the study are going in the direction of the role and importance of multitasking in RTS games, it is necessary to first, extend on the literature of multitasking and second, support these arguments with evidence. So far this section still not strongly supported by empirical evidence and lacks theoretical accounts that can help to justify this link with video games and esports.

Response: Thank you for this comment. Multitasking has been replaced with task-switching. Additionally, we failed to find evidence supporting our arguments (superior task-switching ability could be the key element underlying higher performance in RTS games). Thus, this paragraph has been entirely reworked and the argument about higher performance in RTS games has been deleted.

Change: [P3L45-56] The real-time strategy (RTS) game is a popular esports genre. In an RTS game, players are required to task-switch using complex strategies, because multiple information streams appear simultaneously in different locations on the monitor screen, and players must absorb and select information of importance to them. Based on the information obtained, they must judge how to make an appropriately timed response to the stimuli. It is probable that considering these characteristics of an RTS game, positive effects of playing RTS games on cognitive processes (reaction time and problem-solving ability) occur [10]. Dobrowolski et al. [4] found that RTS players have superior cognitive function compared to that of AVG players. Training with RTS games in elderly people improved task-switching [18]. This could be accompanied by or be based on improving gaze control ability which RTS requires; that is, for smooth task-switching, it is necessary to quickly process multiple sources of visual stimulation.

In light of the above background, the current study aimed to clarify how RTS players control their gaze while playing a game.

P 3 L58-61: The hypothesis does not seem to have a theoretical link still justifying why authors are expecting those findings. However, this hypothesis reads very simplistic, as it is well known that experts will outperform novices in specific tasks. Thus, the hypothesis in its current state does not add knowledge and still too vague.

Response: Thank you for your comment. We reworked the hypothesis to interrelate the theory and previous findings. To be specific, we added some reasons why task-switching ability and gaze control strategy were responsible for the high performance in the RTS game. We believe that it helps to make our hypothesis convincing.

Change: [P3L55-P4L65] In light of the above background, the current study aimed to clarify how RTS players control their gaze while playing a game. In order to shed light on their gaze control strategy, we adopted a popular RTS game, StarCraft, as the model task. Moreover, since task difficulty (i.e., number of tasks that players have to switch between) would be a critical factor for gaze control, it is necessary to clarify whether or not gaze control, while playing an RTS game, depends on task difficulty. Thus, we set the test games with the three different levels of difficulty. Considering the characteristics of RTS games (StarCraft in this case) requiring players to quickly move their gaze over multiple areas on the monitor, and the fact that long-term training in RTS games improves task-switching ability [18], elite StarCraft players would use more ballistic gaze movements, called saccades, to quickly process multiple stimuli. Thus, we hypothesized that RTS experts would show superior gaze control (dispersed and fast saccadic gaze movement) while playing games compared to the novices. We tested this hypothesis by measuring gaze distribution and saccadic movements. 

P3 L60-61: The practical implications as mentioned here for esports coaching and expanding esports markets should be either supported in the introduction with theoretical justifications or addressed at the end of the manuscript supported by the findings.

Response: Thank you for your consideration. According to your comment, we have deleted the descriptions of esports markets. Further, the description of practical applications has been moved to the end of the introduction section with theoretical justifications.

Change: [P22L392-393] Additionally, knowledge about the gaze control strategy used by successful esports players would be helpful for esports coaching and for devising new training methods.

P3 L66-67: As the test seems to fulfil some elements of ecological validity, I would expect justification why multitasking and the increase of task difficulty can help to understand closer what the StarCraft players experience while playing the game. Consequently, support from ecological validity and representative designs studies could help to support the decisions on the difficulty of the task.

Response: Thank you for your suggestion. We explained the reason why task-switching was linked to task difficulty. However, the level of difficulty required to clarify the difference between an expert player and a lower skill player is unknown. Thus, we believe that it is important to explain the different requirement of each task.

Change: [P3L57-P4L59] Moreover, since task difficulty (i.e., number of tasks that players have to switch between) would be a critical factor for gaze control, it is necessary to clarify whether or not gaze control, while playing an RTS game, depends on task difficulty.

P3 L62-64: The investigation used as support for the current study Murphy and Spencer (2009) was a replication study from Green and Bavelier (2003) in which the sample were not esports players as you mention and the games, they are focusing on are not RTS but action games in which combines different games. Thus, this argumentation loses strength and validity when the full story is not explained. The aim of these previous studies should mention and fully considered when comparing the results, aiming to extends your research based on those findings.

Response: Thank you for your comment. We reworked this paragraph by using an RTS game characteristic. When playing an RTS game, players are required to task-switch using complex strategies, because multiple information streams appear simultaneously in different locations on the monitor screen. Long-term training of playing RTS can improve cognitive processing and task-switching ability. However, the gaze control strategy in RTS game players during an actual RTS game has never been investigated. This information (the characteristics of RTS games and the lack of knowledge of the gaze control strategy during an actual RTS game) will help to support our hypothesis.

Change: [P3L45-56] The real-time strategy (RTS) game is a popular esports genre. In an RTS game, players are required to task-switch using complex strategies, because multiple information streams appear simultaneously in different locations on the monitor screen, and players must absorb and select information of importance to them. Based on the information obtained, they must judge how to make an appropriately timed response to the stimuli. It is probable that considering these characteristics of an RTS game, positive effects of playing RTS games on cognitive processes (reaction time and problem-solving ability) occur [10]. Dobrowolski et al. [4] found that RTS players have superior cognitive function compared to that of AVG players. Training with RTS games in elderly people improved task-switching [18]. This could be accompanied by or be based on improving the gaze control ability which RTS requires; that is, for smooth task-switching, it is necessary to quickly process multiple sources of visual stimulation.

In light of the above background, the current study aimed to clarify how RTS players control their gaze while playing a game.

P3 L64-67: I am not sure if this is adding to the hypothesis. It seems that the authors are interested in investigating multitasking and coping mechanisms. The multitasking bit was mentioned earlier but the introduction lacks literature in this field and its importance for the study.

P5 L75: Please be consistent with the language or make sure that connection between gaze control and multitasking is clear. So far, it is confusing to understand which are/is the real aim of the study, as there are different concepts drop during the introduction and methodology. Thus, the theoretical account is not clear, and hypothesis are not supported theoretically or empirically.

Response to the above two comments: Thank you for this consideration. Multitasking has been changed to task-switching ability in this revision. In this paper, we directly focused on the gaze control strategy in RTS experts, rather than the task-switching. Superior gaze control in RTS experts was expected, because the task-switching ability, which is required for high RTS performance, would be supported by gaze control ability. In the revised manuscript, we cited the research which focused on task-switching ability to support our idea (Basak et al. 2008). They argue that playing an RTS game improves the task-switching ability of older adults. Thus, there is a possibility that task-switching ability is an important factor for RTS game players. However, whether gaze control ability supports task-switching ability has never been investigated. This is the reason why we focused on gaze behavior in the current study.

Change: [P4L60-65] Considering the characteristics of RTS games (StarCraft in this case) requiring players to quickly move their gaze over multiple areas on the monitor, and the fact that long-term training in RTS games improves task-switching ability [18], elite StarCraft players would use more ballistic gaze movements, called saccades, to quickly process multiple stimuli. Thus, we hypothesized that RTS experts would show superior gaze control (dispersed and fast saccadic gaze movement) while playing games compared to the novices. We tested this hypothesis by measuring gaze distribution and saccadic movements. 

P5 L78-81: Which preliminary experimental data are you referring to? This needs to be further justified.

Response: We apologize for the missing information. In this paragraph, G*Power has been recalculated by using 4 Expert and 4 Low Skill participants’ performance level data from our preliminary investigation.

Change: [P4L72-P5L77] We performed a power analysis to estimate the required sample size (G*Power version 3.1). The G*Power was calculated using the Hard Task performance level (described in the Sample games section) of experts (Expert, n = 4) and the performance of players with lower skills (Low Skill, n = 4) in a preliminary experiment (Cohen’s d: 1.91; α level: 0.05; power (1-β error probability): 0.9). Effect size was calculated according to Cohen [17]. The results showed that seven participants were required for each group. The details of the G*Power protocol are presented in the S6 Supplement file.

P5 L84-91: The are still information missing. For example, the recruitment strategy and the inclusion/exclusion criteria are still missing.

Response: Thank you for your comment. Information about recruitment strategy and the inclusion/exclusion criteria have been added.

Change: [P4L79-80] Participants were recruited through the Waseda University school bulletin board. Subjects had no record of visual disorders, and were excluded if they had less than six months experience playing StarCraft.

P9 L152-153: I think that the authors should allocate a section where the procedure of the experiment is explained in detailed. Additionally, a figure could help to show the overview of the experimental procedure and measures. This, as I see a critical issue in terms of the procedure of the study as there was not randomization of the difficulty of the tasks.

Response: Thank you for your consideration. We have added a procedure section and a figure showing the overview to our manuscript. We believe that this section will make it easy for readers to understand the experimental procedure as well as the existence of a limitation regarding non-randomization of the difficulty of the tasks. 

Change: [P5L90-96] Experimental procedure

Before the experiment, we measured the distance at 40 inches between the subject’s head and the monitor and asked subjects to maintain the position during the task. Each task was performed for 3 minutes. When the participant failed to play the task for 3 minutes, the task was restarted. Additionally, any task which was not played for 3 minutes was excluded from the analysis. During each task, gaze movement was recorded by an eye tracker (Pupil Core, Pupil Labs). The tasks proceeded in the following order: Easy Task, Moderate Task, and Hard Task. When a task was performed for 3 minutes, the task was over.

Change: [P6L103] Figure 1. Overall flow of the task. Each step shows the overall flow of the task.

P10 L167-169: The preliminary investigations that you are referring to support the sample size and the reliability and validity of the task need to be further described and adequately referenced.

Response: Thank you for this consideration. In the preliminary investigation, four Expert and four Low Skill samples were used to estimate the required sample size. Also, the performance level of the Hard Task in the preliminary investigation was used to estimate the sample size. Specific data about the required sample size was as follows:

Raw data of each participant: Expert (224, 212, 153, 225), Low Skill (146, 176, 113,141)

Power calculation for unpaired t-test between Expert (n = 4) vs. Low Skill (n = 4)

t test: Means: Difference between two independent means (two groups)

Analysis: Compute required sample size

Input: Tail(s) = Two

 Effect size d = 1.91

 α err prob = 0.05

 Power (1-β err prob)) = 0.9

 Allocation ration N2/N1 = 1

Output: Noncentrality parameter δ = 3.57

 Critical t = 2.17

 Df = 12

 Sample size group 1 = 7

 Sample size group 2 = 7

 Total sample size = 14

 Actual power = 0.905

P11 L189: How did you control for the use of different settings of keyboard and mouse? And how this decision could influence the results of the test.

Response: Thank you for the comment. Each participant used their own keyboard and mouse. However, all keyboard and mouse settings were the same for all subjects. Thus, we believe that this difference did not influence the results of experiment.

---

## [Decision Letter · Decision Letter 2]

24 Jan 2022

PONE-D-21-09912R2Difference in gaze control ability between low and high skill players of a real-time strategy game in esportsPLOS ONE

Dear Dr. Nakagawa,

Thank you for submitting your manuscript to PLOS ONE. After careful consideration, we feel that it has merit but does not fully meet PLOS ONE’s publication criteria as it currently stands. Therefore, we invite you to submit a revised version of the manuscript that addresses the points raised during the review process.

We look forward to receiving your revised manuscript.

Kind regards,

Greg Wood, PhD

Academic Editor

PLOS ONE

Reviewers' comments:

Reviewer's Responses to Questions

**Comments to the Author**

1. If the authors have adequately addressed your comments raised in a previous round of review and you feel that this manuscript is now acceptable for publication, you may indicate that here to bypass the “Comments to the Author” section, enter your conflict of interest statement in the “Confidential to Editor” section, and submit your "Accept" recommendation.

Reviewer #1: All comments have been addressed

Reviewer #2: (No Response)

2. Is the manuscript technically sound, and do the data support the conclusions?

Reviewer #1: Yes

Reviewer #2: Partly

3. Has the statistical analysis been performed appropriately and rigorously? 

Reviewer #1: Yes

Reviewer #2: Yes

4. Have the authors made all data underlying the findings in their manuscript fully available?

Reviewer #1: No

Reviewer #2: No

5. Is the manuscript presented in an intelligible fashion and written in standard English?

Reviewer #1: No

Reviewer #2: Yes

6. Review Comments to the Author

Reviewer #1: The authors should be commended for their work in addressing the comments. The manuscript is now ready for publication and I have recommended the manuscript be accepted pending a few small comments.

Firstly on line 72-73, GPower is a software and thus is not calculated. Please amend this sentence.

Secondly, and most importantly the grammar throughout the manuscript is still very poor and the authors are strongly encouraged to have a native english speaker review and revise the manuscript for technical grammar. This includes but is not limited to poor use of plural vs non plural words in wrong instances and the use of past and present tenses in the same sentence. Having been immersed in the review of the manuscript, I found myself able to adapt to these shortcomings to review the scientific merit of the work, but I fear the general readership will have trouble with the readability of the manuscript in its current state.

Reviewer #2: Comments to the authors

Thank you for resubmitting your manuscript and consideration of my comments.

Overall, there are improvements to the manuscript and more consistency in language and justification of specific decisions. However,

P2 L34-36: Please use the current literature to justify concepts and definitions (see Pedraza-Ramirez et al., 2020).

P2 L37-40: Please include systematic reviews and/or meta-analysis to justify the interest in researching the cognitive elements of esports, rather than highlighting that there are cognitive benefits already seen by the reference used of 2018. This seems to be misleading as only until 2020 and 2021 did research in esports cognition start to be published. To see some examples that can serve as supporting references for video games cognition and esports cognition please see accordingly (Pedraza-Ramirez et al., 2020; Powers et al., 2013).

P3 L55-57: The aim to use the esports StarCraft does not mean that you can generalise the investigation to other RTS games. This issue can be seen in studies investigating cognitive function in action video games or in more specific genres like MOBAs (i.e., League of Legends, Dota) where mixed evidence has been found (e.g., Boot et al., 2008, Kokkinakis et al., 2017), by generalising findings of the investigation in one esports to the whole game genre. Therefore, I recommend clarifying and keeping it consistent through the manuscript that the specific cognitive process is closely related to the demands of the esport under investigation without generalising the findings to other similar esport games. This, as supported and suggested by different authors in the video game and esports research (Power et al., 2013; Dale & Green, 2017; Pedraza-Ramirez et al., 2020).

P4 L61-64: This claim shown as fact cannot be done by one individual study in an elderly population, additionally the study is from 2008 with very specific conditions. Therefore, the theoretical support for the hypothesis is still missing as currently there is no supporting evidence that justifies the expected expertise differences. Thus, I recommend you search in the recent reviews and empirical studies.

Suggested References:

Boot, W. R., Kramer, A. F., Simons, D. J., Fabiani, M., & Gratton, G. (2008). The effects of video game playing on attention, memory, and executive control. ACTA PSYCHOLOGICA, 129(3), 387-398. doi:10.1016/j.actpsy.2008.09.005

Dale, G., & Green, C. S. (2017). The Changing Face of Video Games and Video Gamers: Future Directions in the Scientific Study of Video Game Play and Cognitive Performance. Journal of Cognitive Enhancement, 1(3), 280-294. doi:10.1007/s41465-017-0015-6

Kokkinakis, A. V., Cowling, P. I., Drachen, A., & Wade, A. R. (2017). Exploring the relationship between video game expertise and fluid intelligence. PLOS ONE, 12(11). doi:10.1371/journal.pone.0186621

Pedraza-Ramirez, I., Musculus, L., Raab, M., & Laborde, S. (2020). Setting the scientific stage for esports psychology: a systematic review. International Review of Sport and Exercise Psychology, 0(0), 1-34. doi:10.1080/1750984X.2020.1723122

Powers, K. L., Brooks, P. J., Aldrich, N. J., Palladino, M. A., & Alfieri, L. (2013). Effects of video-game play on information processing: A meta-analytic investigation. Psychonomic Bulletin and Review, 20(6), 1055-1079. doi:10.3758/s13423-013-0418-z

7. PLOS authors have the option to publish the peer review history of their article (what does this mean?). If published, this will include your full peer review and any attached files.

Reviewer #1: No

Reviewer #2: No

---

## [Author Response · Author response to Decision Letter 2]

21 Feb 2022

Dear Reviewers

Thank you very much for your feedback, comments, and suggestions. Our responses to your comments are as follows:

Reviewer #1:

The authors should be commended for their work in addressing the comments. The manuscript is now ready for publication and I have recommended the manuscript be accepted pending a few small comments.

Response: Thank you for your general comments. Your comments were very helpful to improve

our manuscript.

Firstly on line 72-73, GPower is a software and thus is not calculated. Please amend this sentence.

Response: Thank you for your comment. We have amended this sentence in response to your comment.

Change: [P5L74-77] The power analysis was conducted using the Hard Task performance level (described in the Sample games section) of experts (Expert, n = 4) and the performance of players with lower skills (Low Skill, n = 4) in a preliminary experiment (Cohen’s d: 1.91; α level: 0.05; power (1-β error probability): 0.9).

　

Secondly, and most importantly the grammar throughout the manuscript is still very poor and the authors are strongly encouraged to have a native English speaker review and revise the manuscript for technical grammar. This includes but is not limited to poor use of plural vs nonplural words in wrong instances and the use of past and present tenses in the same sentence. Having been immersed in the review of the manuscript, I found myself able to adapt to these shortcomings to review the scientific merit of the work, but I fear the general readership will have trouble with the readability of the manuscript in its current state.

Response: Thank you for your suggestion. According to your suggestion, the manuscript has been proofread by a native English speaker who is a professional scientific editor (as mentioned in the Acknowledgment section).

Reviewer #2:

Thank you for resubmitting your manuscript and consideration of my comments.

Overall, there are improvements to the manuscript and more consistency in language and justification of specific decisions. However,

P2 L34-36: Please use the current literature to justify concepts and definitions (see Pedraza-Ramirez et al., 2020).

Response: Thank you for your suggestion. We added the reference which you suggest.

Change: [P2L34-35] Today it is common to play a competitive game with other players online; this activity is called electronic sports (esports) [1].

P2 L37-40: Please include systematic reviews and/or meta-analysis to justify the interest in researching the cognitive elements of esports, rather than highlighting that there are cognitive benefits already seen by the reference used of 2018. This seems to be misleading as only until 2020 and 2021 did research in esports cognition start to be published. To see some examples that can serve as supporting references for video games cognition and esports cognition please see accordingly (Pedraza-Ramirez et al., 2020; Powers et al., 2013).

Response: Thank you for your suggestion. We have cited some meta-analysis research according to your suggestion. 

Change: [P2L38-P3L41] In addition, meta-analysis revealed that playing a commercial video game improves the information-processing skills of players, which include task-switching ability and visual processing [1, 3]. Task-switching is defined as the ability to quickly alternate among multiple separate tasks [4].

P3 L55-57: The aim to use the esports StarCraft does not mean that you can generalise the investigation to other RTS games. This issue can be seen in studies investigating cognitive function in action video games or in more specific genres like MOBAs (i.e., League of Legends, Dota) where mixed evidence has been found (e.g., Boot et al., 2008, Kokkinakis et al., 2017), by generalising findings of the investigation in one esports to the whole game genre. Therefore, I recommend clarifying and keeping it consistent through the manuscript that the specific cognitive process is closely related to the demands of the esport under investigation without generalising the findings to other similar esport games. This, as supported and suggested by different authors in the video game and esports research (Power et al., 2013; Dale & Green, 2017; Pedraza-Ramirez et al., 2020).

Response: Thank you for your opinion. We modified the sentences to emphasize that our aim was directed towards StarCraft players without generalizing the findings to other esports games. Not only P3L56-57 but also in the entire Introduction and Discussion sections, we have changed the description that formerly generalized the current finding to all esports. Alternatively, we added a sentence regarding the possibility that our findings (StarCraft players have a specific gaze control strategy) may be generalizable to other similar esports games. 

Change: [P3L56-57] In light of the above background, we used a popular RTS game, StarCraft, as a research model of RTS, and aimed to clarify how StarCraft experts control their gaze during their play to estimate their gaze control ability.

Change: [P22L393-396] Additionally, Dale and Green discovered that training using RTS games improves task-switching ability [12]. This suggests that our findings (StarCraft experts have a specific gaze control strategy) can be generalized to other esports games, and we believe that the specific gaze control strategy used by expert esports players is the source of their superior performance.

P4 L61-64: This claim shown as fact cannot be done by one individual study in an elderly population, additionally the study is from 2008 with very specific conditions. Therefore, the theoretical support for the hypothesis is still missing as currently there is no supporting evidence that justifies the expected expertise differences. Thus, I recommend you search in the recent reviews and empirical studies.

Response: Thank you for your suggestion. We have added the following references. Chiappe et al. (2013) argue that a first-person shooting (FPS) game improves task-switching ability, and according to Klaffehn et al. (2018), FPS players and RTS players have similar task-switching ability. To be specific, RTS players and FPS players have similar reaction times during task switching. Finally, not only a single case study, but also a meta-analysis suggests that RTS improves task-switching ability (Dale and Green, 2017). We believe these references provide supporting evidence that justifies the expected expertise differences. 

Change: [P4L60-64] Additionally, it is already known that first-person shooting (FPS) games improve task-switching ability [14]; in addition, not only a single case study but also a meta-analysis study argues that RTS improves task-switching ability [11]. One study has found that FPS games and RTS games have a similar positive effect on task-switching ability [15]. Thus, elite StarCraft players should have superior task-switching ability. 

[References] 

11. Dale G, Shawn Green C. The Changing Face of Video Games and Video Gamers: Future Directions in the Scientific Study of Video Game Play and Cognitive Performance. J Cogn Enhanc [Internet]. 2017;1(3):280–94. Available from: https://doi.org/10.1007/s41465-017-0015-6

14. Chiappe D, Conger M, Liao J, Caldwell JL, Vu K-PL. Improving multi-tasking ability through action videogames. Appl Ergon [Internet]. 2013;44(2):278–84. Available from: https://www.sciencedirect.com/science/article/pii/S0003687012001202

15. Klaffehn AL, Schwarz KA, Kunde W, Pfister R. Similar Task-Switching Performance of Real-Time Strategy and First-Person Shooter Players: Implications for Cognitive Training. J Cogn Enhanc [Internet]. 2018;2(3):240–58. Available from: https://doi.org/10.1007/s41465-018-0066-3

---

## [Editor Report · Decision Letter 3]

4 Mar 2022

Difference in gaze control ability between low and high skill players of a real-time strategy game in esports

PONE-D-21-09912R3

Dear Dr. Nakagawa,

We’re pleased to inform you that your manuscript has been judged scientifically suitable for publication and will be formally accepted for publication once it meets all outstanding technical requirements.

Kind regards,

Greg Wood, PhD

Academic Editor

PLOS ONE
---

## [Editor Report · Acceptance letter]

10 Mar 2022

PONE-D-21-09912R3 

Difference in gaze control ability between low and high skill players of a real-time strategy game in esports 

Dear Dr. Nakagawa:

I'm pleased to inform you that your manuscript has been deemed suitable for publication in PLOS ONE. Congratulations! Your manuscript is now with our production department. 

Kind regards, 

on behalf of

Dr. Greg Wood 

Academic Editor

PLOS ONE